# Scavenger receptor CD163 multimerises to allow uptake of diverse ligands

Richard X. Zhou [1,2] & Matthew K. Higgins [1,2] ✉

CD163 is an archetypal scavenger receptor and mediates detoxification of free haemoglobin. Release of haemoglobin from lysed erythrocytes causes oxidative tissue and organ damage. Detoxification involves haemoglobin binding to the abundant serum protein haptoglobin, followed by CD163-mediated uptake of stoichiometrically diverse haptoglobin-haemoglobin complexes into macrophages for degradation. We show that CD163 adopts dimeric and trimeric assemblies due to calcium-mediated interactions within a membrane-associated base. Arms protrude from this base and create a ligand-binding site. Flexibility within the base, coupled with multiple small ligand-binding surfaces on each arm, allow the receptor to mould around its ligands, resulting in promiscuous uptake of ligands with different structures and stoichiometries. Monomeric CD163 lacks this ability to internalise lower-avidity ligands. Arms from adjacent protomers can also self-associate, blocking ligand-binding surfaces in an autoinhibited state. Therefore, through calcium-dependent multimer formation and flexible ligand binding, CD163 scavenges ligands with different structures and avidities, mediating haemoglobin detoxification.

Transport of oxygen in mammalian blood is mediated by haemoglobin (Hb) tetramers, which circulate within erythrocytes. But free Hb is toxic, due to the reactive properties of the haem group, which can engage in oxidative reactions and generate free radicals[1–3]. Hb can be released from erythrocytes due to haemolysis or tissue damage, and free Hb increases in chronic conditions such as sickle cell anaemia[4] or during infections such as malaria[5]. Mammals have therefore evolved a system to detoxify free Hb.

Hb detoxification involves the abundant serum protein haptoglobin (Hp). The serine protease (SP) domain of Hp binds to an Hb αβ-dimer[6,7]. Hp also contains complement control protein (CCP) domains, which differ in number in the two isoforms found in humans[8]. In people homozygous for isoform 1, a single CCP domain per Hp chain mediates dimer formation, creating a 'dumbbell'-shaped haptoglobin-haemoglobin (HpHb) complex with two 'heads'. In contrast, isoform 2 allows formation of higher-order oligomeric states in individuals who are either homozygous for isoform 2 or heterozygous[9–11]. The formation of HpHb complexes buries reactive groups on Hb and reduces their capacity to cause oxidative damage[6].

The second step in detoxification is the uptake of HpHb into macrophages. This is mediated by CD163[12], which is the best-characterised member of the class I scavenger receptor family. CD163 contains an ectodomain consisting of nine scavenger receptor cysteine-rich (SRCR) domains linked to a C-terminal transmembrane helix[13]. The CD163 binding site has been localised to SRCR1-5[14], and mutation of two acidic amino acid residues in SRCR2 and 3 disrupts binding[15], as does mutation of two positively charged side chains in Hp[15]. A similar acidic cluster was previously observed in the SRCR domain of the scavenger receptor MARCO, which binds to calcium[16]. Indeed, HpHb binding by CD163 is also calcium-dependent, with high-affinity binding in physiological calcium concentrations and release in low calcium conditions[14]. Indeed, the low calcium concentration and reduced pH in the endosomes has been proposed to cause HpHb release to allow degradation[12,14,17].

To mediate its physiological function, CD163 must recognise and take up different ligands. These include HpHb complexes with multiple molecular architectures, resulting from combinations of different Hp isoforms, all of which bind CD163[10,12]. In addition, it

[1]Department of Biochemistry, University of Oxford, South Parks Road, Oxford OX1 3QU, UK. [2]Kavli Institute for Nanoscience Discovery, Dorothy Crowfoot Hodgkin Building, University of Oxford, South Parks Road, Oxford OX1 3QU, UK. ✉e-mail: matthew.higgins@bioch.ox.ac.uk

is disadvantageous for CD163 to internalise free Hp as this would deplete serum Hp without detoxification of Hb. Whether uptake of Hb tetramers occurs is less clear, with no binding of CD163 to Hb observed in one study[10], but with CD163-mediated uptake shown in another, albeit less efficiently than that of HpHb[18]. In a healthy individual, Hb binding is not likely to be required as Hp will rapidly assemble into HpHb complexes. However, in cases of ahaptoglobinaemia, when Hp is depleted during infection, such as malaria[5], or in individuals with genetic polymorphisms such as sickle cell disease[4], free Hb will accumulate and Hb binding and uptake might be advantageous.

While the structure of isoform 1 HpHb complex has been determined[6,19], little was known about how CD163 selectively binds to a structurally diverse range of HpHb complexes, why it does not bind to Hp and how it releases its ligand. We therefore combined cryogenic electron microscopy, biophysics and cell-based uptake assays to understand the molecular mechanism of CD163-mediated ligand uptake and release.

## Results

### The structural basis for binding of haptoglobin-haemoglobin to CD163

To understand how human CD163 binds to HpHb, we expressed the complete ectodomain in HEK293 cells. We then assembled a complex of CD163 with human HpHb, using Hp isoform 1, as, unlike isoform 2, this forms a homogeneous dumbbell-shaped complex, consisting of two haptoglobin molecules, each bound to a haemoglobin dimer[6], which we hypothesised would be well-ordered for structural studies. Hp(1-1)Hb complexes were assembled by mixing human Hp isoform 1 with human Hb and were then combined with CD163 ectodomain in the presence of 2.5 mM $CaCl_2$, to match the calcium concentration found in serum and ensure calcium saturation of the receptor (Supplementary Fig. 1a)[14]. As the stoichiometry was unclear at the time, we used a molar excess of Hp(1-1)Hb, assuming one Hp(1-1)Hb dimer to bind to two CD163 receptors, as previously proposed[6]. The complex was purified by size-exclusion chromatography, and grids were prepared for cryogenic electron microscopy (Supplementary Fig. 1).

Data were collected on a Titan Krios and processed using SIMPLE[20] and CryoSPARC[21] (Supplementary Figs. 2, 4, 5 and Supplementary Table 1), resulting in two predominant three-dimensional classes. In both cases, we observed a single SP domain of Hp bound to an Hb αβ-dimer, and this was coordinated by either two or three CD163 ectodomains (Fig. 1a, b). We did not observe clear density for the second half of the Hp(1-1)Hb dumbbell previously observed in a crystal structure[6], although weaker density was observed for the CCP domain of Hp, positioned correctly relative to the SP domain. Why the second half of the dumbbell was not seen is not clear, but may result from flexibility in the linkage between SP and CCP domains, which may have been rigidified in the previous structure due to crystal packing. Indeed, the crystal structure of Hp(1-1)Hb shows only a small contact area between the SP and CCP domains[6]. Size-exclusion chromatography with multi-angle laser light scattering (SEC-MALLS) supports our structural data, showing concentration-dependent multimer formation in solution, with the addition of HpHb increasing the fraction of multimeric complexes. However, at the concentrations tested, a dimer of HpHb with two receptor trimers was not seen (Supplementary Fig. 1b, c).

The CD163 ectodomain adopts an architecture consisting of a 'base', formed from SRCR domains 5–9, linked to a four-domain 'arm' (SRCR1-4), of which two or three domains were ordered (Fig. 1c). The five domains of the base form a compact assembly which folds to bring SRCR5 and SRCR9 from the same monomer together. In both the dimer and trimer, multimer formation is mediated by the base, predominantly due to interactions between SRCR7 of one subunit and SRCR9 of its neighbour (Fig. 1d). These interfaces contain spheres of density, which we attribute to be calcium ions, as calcium is required

for HpHb binding and uptake[12,14]. The three interfaces within the trimer differ, with 'rocking' motions around these calcium ions leading to different relative positions of the two protomers (Fig. 1d).

The CD163 trimer has a flat triangular base, with the C-termini, which link to transmembrane helices, emerging from the flat surface (Fig. 1a). Protruding in the opposite direction are the arms, which extend away from the membrane. These three arms collectively form a binding site for one head of HpHb, with each arm interacting with a different surface of the head (Supplementary Table 3). The dimer adopts a similar global architecture, with the arm of chain A of CD163 ($CD163_A$) making the same contacts with HpHb as in the trimer, but with the second arm making different interactions. While each arm adopts a very similar conformation (Fig. 1c), flexibility in their relative positions occurs due to rocking motion within the base (Fig. 1e). This allows the arms to emerge from the base at different angles and to make different contacts with the asymmetric HpHb head. Flexibility of the CD163 base, therefore, allows its arms to mould to the ligand, likely allowing a scavenger receptor to create different binding sites for differently structured ligands.

### Structures of unliganded CD163 suggest a mechanism of autoinhibition

To determine whether the HpHb binding site is pre-formed in the absence of ligand, we also used cryogenic electron microscopy to reveal the structure of unliganded CD163. In this case, particles were distributed into three distinct three-dimensional classes (Supplementary Figs. 3–5 and Supplementary Table 2). The first class showed a trimeric arrangement of CD163 (Fig. 2a). Here, the base was well-resolved, adopting a similar conformation to that observed in the presence of ligand. In contrast, the density for the arms was poorly defined.

The other two three-dimensional classes revealed dimeric and trimeric arrangements of the unliganded receptor in which the arms of neighbouring protomers form contacts with each other. These classes were not seen in the presence of the ligand, suggesting that the ligand can compete for the formation of these interactions. The base, formed from SRCR5-9, adopted a similar arrangement in the unliganded dimer to that seen in the liganded dimer (Fig. 2b and Supplementary Fig. 6). However, the arms were in different conformations. In the dimeric unliganded form, SRCR2 and SRCR3 of $CD163_A$ interact with SRCR3 and SRCR4 of $CD163_B$ (Fig. 2b, c and Supplementary Table 3). In the trimeric unliganded form, the same contacts between $CD163_A$ and $CD163_B$ are observed, with additional weaker density for the arm of $CD163_C$, showing it to contact the arms of the other two subunits (Supplementary Fig. 6). The surfaces of the SRCR domains which mediate contacts between the arms overlap with those which interact with HpHb (Fig. 2d). Contacts between arms will therefore compete with ligand-binding interactions and the arms must move apart to create a ligand-binding site.

Unliganded CD163 on the surface of macrophages is therefore likely to exist in an equilibrium in which the arms are either available for ligand binding or interact together, occluding ligand binding. Interactions between arm and ligand will be in competition with interactions between neighbouring arms, providing a mechanism of autoinhibition. PISA analysis shows that the arm-arm interactions have a total interface area of ~590 Å², while the dimeric and trimeric receptors have total contact areas with HpHb of ~930 and ~1445 Å², suggesting that HpHb will effectively compete with these arm-arm interactions and explaining why we do not see these auto-inhibited CD163 conformations in the presence of ligand. In contrast, weakly binding ligands with smaller binding interfaces may not be able to compete and will not be internalised. These structural data are therefore consistent with a model in which arm-arm interactions might outcompete the binding of weak ligands, but HpHb will outcompete the arm-arm interactions, allowing HpHb uptake.

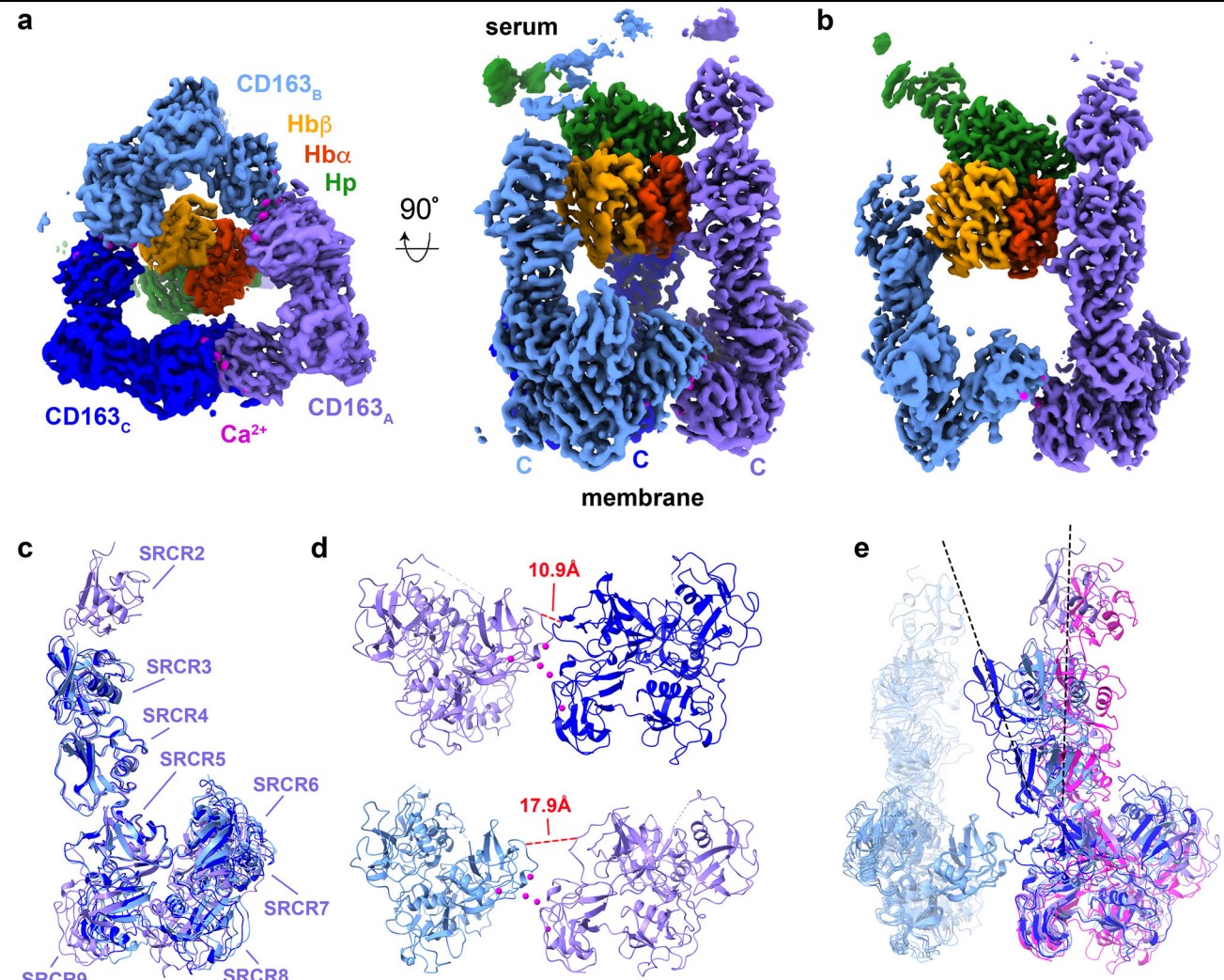

**Fig. 1 | The structure of CD163 bound to HpHb. a** The structure of a trimer of the CD163 ectodomain bound to the HpHb complex. The three copies of CD163 are shown in three shades of blue. HpSP domain is green, the α-subunit of Hb is red and the β-subunit of Hb is orange. **b** The structure of a dimer of the CD163 ectodomain bound to the HpHb complex, coloured as in (**a**). **c** An alignment of the three CD163 molecules found in the trimeric complex, aligned on SRCR5, with the SRCR domains labelled. **d** A close-up of the interfaces between two pairs of CD163 subunits with the red dotted line showing the different distances between the Cα atoms of residues 561 and 804 in these two interfaces. **e** An alignment of each pair of CD163 ectodomains observed in the trimeric and dimeric complex, with each aligned on SRCR7 of the left-hand CD163. The right-hand CD163 molecules from the trimer are coloured as in (**a**). and in pink for the CD163$_A$ from the dimer. Dashed lines indicate the maximum degree of tilting of the arm of CD163 in these structures.

## CD163 mediates the uptake of ligands with different stoichiometries and structures

Analysis of the HpHb-bound trimer showed that each arm of CD163 interacts differently with HpHb (Fig. 3a and Supplementary Table 3). CD163$_A$ interacts with the Hb α-subunit through SRCR3 and SRCR4, and with Hp through SRCR2. CD163$_B$ recognises the β-subunit of Hb through SRCR3 and SRCR4, and CD163$_C$ binds to the α-subunit of Hb through SRCR3 and SRCR4 and to Hp through SRCR3. Remarkably, it is the same faces of these SRCR domains which mediate different interactions with different surface features of HpHb, with the plasticity of the HpHb-binding site forming an asymmetric binding site, despite being formed from a homotrimer of receptors.

HpHb complexes are found in a variety of multimeric states. The dumbbell-shaped Hp(1-1)Hb is formed when an Hp(1-1) dimer, presenting two protomers of HpSP, binds to an Hb αβ-dimer on each end[6]. In contrast, higher-order multimeric states of Hp are found in people homozygous for isoform 2, leading to formation of Hp(2-2)Hb[11] (Fig. 3b). We observed that the three arms of CD163 come together to interact with a single head of HpHb, with weak density for the CCP

domain of Hp projecting away from the membrane towards the other head (Fig. 1b). This leads to the hypothesis that HpHb complexes formed from different isoforms, and an artificial version containing just the HpSP domain bound to the Hb dimer (HpSPHb), should show similar binding and uptake properties. The structure also showed that ~68% of the CD163 interaction surface on the HpHb complex is mediated by Hb subunits, with ~32% due to Hp, suggesting that CD163 might mediate Hb uptake more efficiently than that of Hp. Additionally, human sera contains haptoglobin-related protein (Hpr)[22], and a complex of Hpr and Hb is part of the trypanolytic factor, which kills African trypanosomes[19,23,24]. It is important that this is not taken up into human macrophages[23], where it might cause cell death. Polymorphisms between Hp and Hpr are found in residues which contact CD163 (Fig. 3c), suggesting that they might prevent binding. To test these predictions, we developed binding and uptake assays and assessed the outcomes for different ligands.

We first assembled Hp(1-1)Hb, Hp(2-2)Hb, HpSPHb and a complex of the SP domain of Hpr bound to Hb (HprSPHb). We coupled the CD163 ectodomain with a biotinylated BAP tag at the C-terminus to a

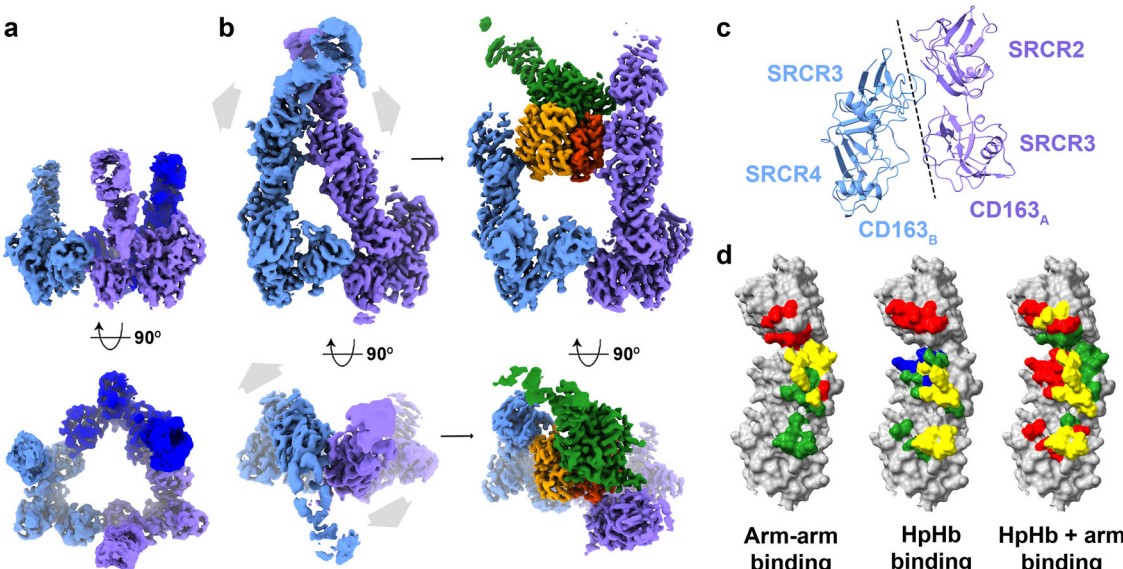

**Fig. 2 | The structure of unliganded CD163. a** The structure of a trimer of CD163 in the absence of ligand, with disordered arms, coloured as in Fig. 1. **b** The structure of a dimer of CD163 in the absence of ligand (left) and in the presence of HpHb (right), showing arm-arm contacts between the CD163 arms in the unliganded dimer, with the arms moving outwards to allow ligand binding. **c** Close-up on the interface between the arms of CD163$_A$ and CD163$_B$ in the unliganded dimer. **d** A schematic presenting the overlap in residues involved in arm-arm contacts and HpHb binding. The left panel shows SRCR2-4 of CD163 as a surface representation with residues involved in arm-arm contacts in the unliganded dimer coloured red for CD163$_A$, green for CD163$_B$ and yellow if involved in both subunits. In the central panel, residues are labelled if directly contacting HpHb in the liganded dimer and trimer. Red represents CD163$_A$, green CD163$_B$, blue CD163$_C$, with yellow depicting residues involved in ligand binding in at least two CD163 subunits. The right-hand panel shows the same surface with residues labelled red if they contact HpHb, green if they are involved in arm-arm contacts in the unliganded dimer and yellow if both.

surface plasmon resonance (SPR) chip surface, ensuring presentation with an orientation which matches that on a membrane. This will result in the conjugation of a mixture of multimeric states, as observed in the cryo-EM data. We then flowed the different HpHb complexes over the surface. Hp(1-1)Hb, Hp(2-2)Hb and HpSPHb bound with apparent dissociation constants of 0.28, 0.16 and 1.3 nM, supporting the hypothesis that CD163 complexes recognise a single HpHb head (Fig. 3d, Supplementary Fig. 7a and Supplementary Table 4). In contrast, HprSPHb showed no binding at a concentration of 20 nM. This assay could not be used to measure the affinity for Hb and Hp, due to the irregular shape of the SPR sensorgrams and to high non-specific binding. We instead used microscale thermophoresis (MST), which revealed affinities of 0.19 µM for Hb and 27 µM for Hp (Fig. 3e, Supplementary Fig. 8 and Supplementary Table 4). Therefore, HpHb binds to CD163 most tightly, followed by Hb, with little binding observed for Hp or HprSPHb.

In addition, we used a cell-based uptake assay in which HEK293 cells were transfected with full-length CD163 modified with a cytoplasmic GFP tag[17]. Here, the Hp(2-2)Hb complex was labelled and its uptake was assessed. Hp(2-2)Hb shows lower non-specific uptake than Hp(1-1)Hb due to its larger size, giving greater specificity to the assay. The inhibition of Hp(2-2)Hb uptake due to competition from different ligands could then be quantified. In this assay, we showed that uptake of fluorescently labelled Hp(2-2)Hb in HEK293 cells increases nearly tenfold in the presence of CD163 (Supplementary Fig. 9a). We then assessed the ability of different concentrations of unlabelled ligands to compete for the uptake of 50 nM labelled Hp(2-2)Hb, as quantified by fluorescence-activated cell sorting (FACS) (Fig. 3f). Our BSA negative control and unconjugated Hp isoform 1 both showed a similar concentration dependence, suggesting no receptor-mediated uptake of Hp at 4 µM concentration. As Hp is normally found in sera at concentrations of 5–30 µM[25], receptor-mediated internalisation will be low. In contrast, Hp(1-1)Hb, Hp(2-2)Hb and HpSPHb all competed with similar IC$_{50}$ values of 33, 40 and 41 nM, while Hb competed with a

greater IC$_{50}$ of 0.25 µM. HprSPHb showed some specific uptake at concentrations above 2 µM, which is greater than the normal serum concentration of ~1 µM[23]. Therefore, uptake experiments and binding studies were consistent, showing that all HpHb variants are equally endocytosed at low concentration, while Hp is not. Free Hb will be endocytosed at higher concentrations, which may be physiologically important in disease or in infection where haemolysis results in Hp depletion.

### Calcium regulates CD163 multimer formation and HpHb uptake

The structure of the CD163 trimer bound to HpHb revealed spheres of density, which we attribute to be calcium ions, present both at the interfaces between the protomers, and at the interface between the arm of CD163$_A$ and HpHb (Fig. 4a). We therefore assessed whether calcium is important for CD163 multimer formation and for HpHb binding and uptake.

We used SEC-MALLS to assess the multimeric state of liganded CD163. In the presence of a physiologically relevant calcium concentration of 2.5 mM CaCl$_2$ at pH 7.5, the ectodomain showed concentration-dependent multimerisation (Fig. 4b and Supplementary Fig. 1c). In contrast, in the absence of calcium, at a lower pH of 6 or when CaCl$_2$ is replaced by 2.5 mM MgCl$_2$, CD163 was monomeric. We also used SPR to test whether the presence of calcium is required for HpHb binding. While Hp(1-1)Hb bound with apparent dissociation constants of 0.28 nM in the presence of calcium (Fig. 3d), no binding was observed in the absence of calcium, in the presence of 2.5 mM MgCl$_2$ or at pH 6 (Fig. 4b and Supplementary Fig. 7b).

Finally, we assessed the importance of calcium for uptake into HEK293 cells expressing CD163. As the media used for these experiments contains ~1.8 mM CaCl$_2$, we performed the assays in the presence of 2 mM EGTA to chelate calcium. While EGTA had no effect on uptake of transferrin (Supplementary Fig. 9b), suggesting that it did not negatively affect global endocytosis, it resulted in a reduction of HpHb uptake to levels observed for cells lacking CD163 (Fig. 4c).

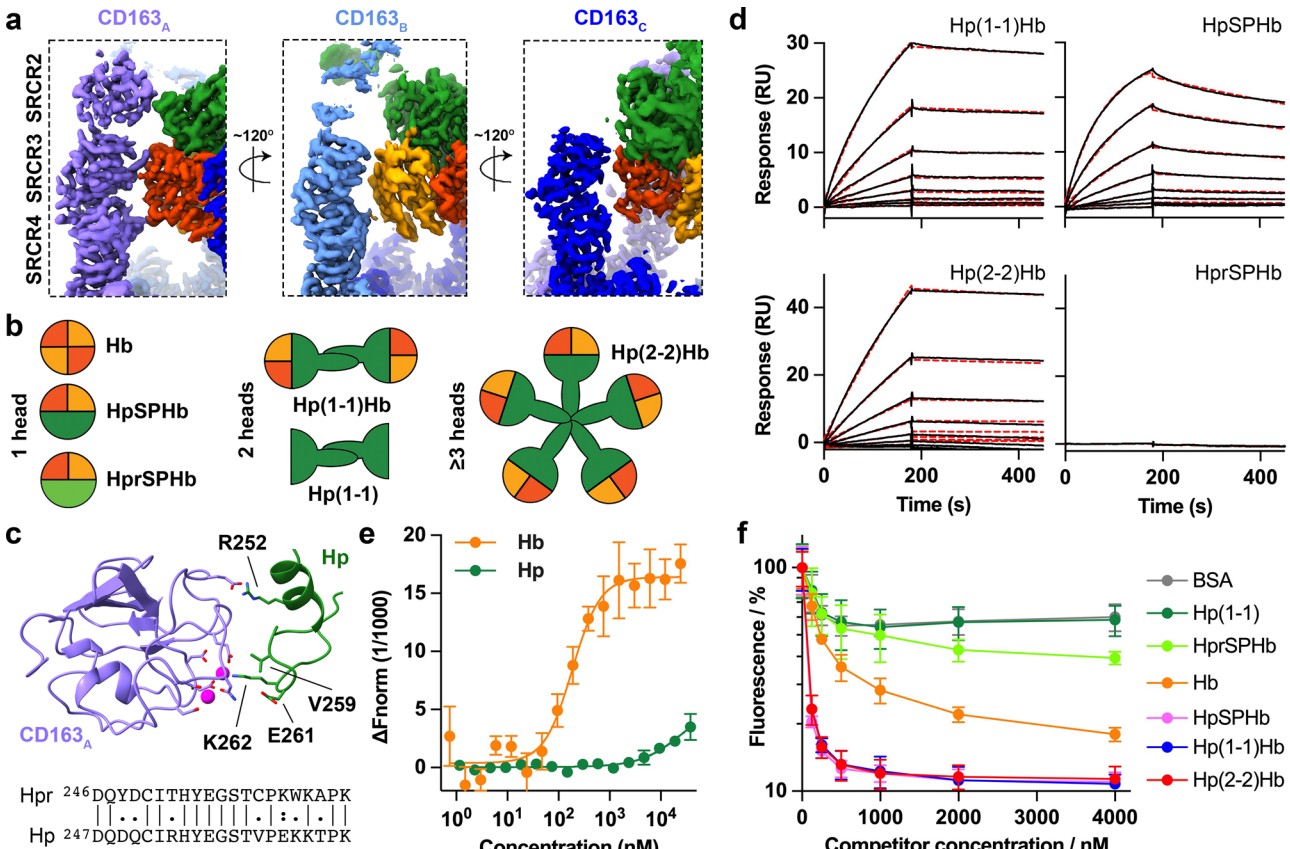

**Fig. 3 | Ligand selectivity of CD163. a** The structure of the CD163 trimer bound to HpHb viewed from three directions, showing that each arm of CD163 interacts differently with the HpHb ligand. **b** Schematic showing the different stoichiometries and structures of the ligands studied here. **c** A view of SRCR2 of CD163$_A$ (violet) bound to Hp (green). Below is shown a partial sequence alignment of Hp and Hpr. Residues which differ between Hp and Hpr and which might influence binding to CD163 are shown as sticks and are labelled. **d** Assessment of the binding of CD163 ectodomain, immobilised through a C-terminal biotin, to Hp(1-1)Hb, Hp(2-2)Hb, HpSPHb and HprSPHb by SPR analysis. These represent twofold dilution series from a maximum concentration of 10 nM for Hp(1-1)Hb and Hp(2-2)Hb, 20 nM for HpSPHb, as well as an injection at 20 nM of HprSPHb. Data were shown as black lines, while fitting to a one-to-one binding model is depicted as dashed red lines. These are representative of $n$ = 3. **e** Assessment of the binding of Hp and Hb to the CD163 ectodomain using MST. Each point is the mean of three replicates, and the error bars are the standard deviation. **f** Measurement of the ability of different ligands to compete for the uptake of fluorescently labelled Hp(2-2)Hb into HEK293 cells transfected with CD163. BSA is included as a control for CD163-independent effects. Each point represents the mean of three replicates, and the error bars are standard deviations.

Therefore, calcium, together with pH, mediates formation of CD163 multimers and is required for HpHb uptake. As both calcium concentration and pH are low in the endosome, this suggests that CD163 forms multimers in the serum, which efficiently take up ligands and that these multimers disassemble in the endosomes, facilitating HpHb release.

## Monomeric CD163 is less effective at the uptake of lower-avidity ligands

Having shown that calcium is important for CD163 multimer formation, we next aimed to determine whether the putative calcium ions at the binding interface between CD163 and HpHb are also important. To achieve this, we developed a monomeric CD163 molecule. The structure shows that N807 occupies a central location within the multimer interface (Fig. 5a). Therefore, an R809T mutant was produced to convert N807 into an N-linked glycosylation consensus sequence, with the presence of the glycan predicted, based on the structure, to sterically prevent multimer formation. This mutant remained monomeric at a monomer concentration of 24.2 μM, as measured by SEC-MALLS (Fig. 5b). SPR measurements showed that monomeric CD163 bound to Hp(1-1)Hb with an apparent affinity of 1.2 nM in the presence of 2.5 mM CaCl₂ (Fig. 5c, Supplementary Fig. 7a and Supplementary Table 4), compared with 0.28 nM for the native protein (Fig. 3d). This

interaction was abolished in the absence of calcium (Fig. 5c) or when CaCl₂ was replaced with 2.5 mM MgCl₂ (Supplementary Fig. 7b). Therefore, calcium ions are important for the direct interaction of CD163 with its ligands in addition to their role in mediating multimerisation.

The availability of monomeric CD163 also allowed us to assess the degree to which multimer formation is important for binding and uptake of diverse ligands. We first measured the affinities of different ligands for the R809T mutant, using SPR for HpHb complexes and MST for Hp and Hb. We found that apparent affinities for monomeric CD163 differed more than 1000-fold between Hp(2-2)Hb and HpSPHb, with values at 31 pM and 38 nM, respectively (Fig. 5c, Supplementary Fig. 7a and Supplementary Table 4). This stands in contrast to the similar affinities observed between HpHb variants for multimeric CD163 (Fig. 3d). The affinity for Hb was estimated to be 1.9 μM, a tenfold decrease compared to the wild-type receptor and no Hp binding was detected at 96 μM CD163 monomer (Fig. 5d, Supplementary Fig. 8 and Supplementary Table 4. Therefore, monomeric CD163 bound to all ligands, other than Hp(2-2)Hb, with a lower apparent affinity than multimeric CD163.

To assess the impact of the CD163 multimer formation on the uptake of various ligands, we generated a HEK293 cell line in which full-length monomeric R809T was expressed. This allowed us to

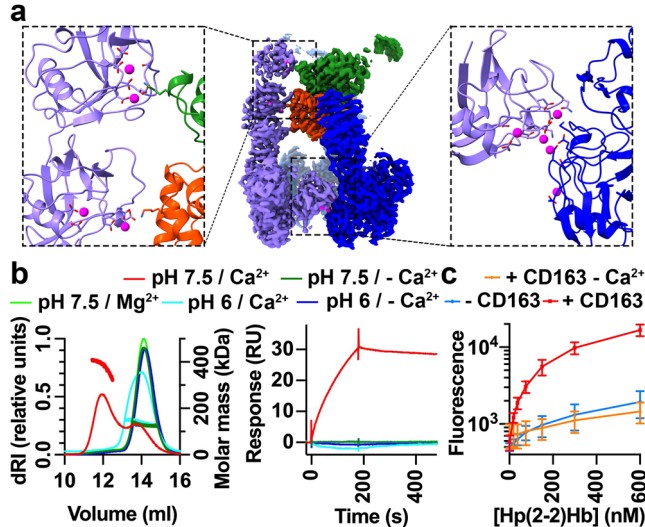

**Fig. 4 | Calcium is required for CD163 multimer formation and for HpHb uptake. a** The central panel shows the structure of the CD163 trimer bound to HpHb, coloured as in Fig. 1. The left-hand panel shows $Ca^{2+}$ ions at the interface between $CD163_A$ and HpHb, while the right-hand panel shows $Ca^{2+}$ ions at the interface between $CD163_A$ and $CD163_C$. **b** The left-hand panel shows SEC-MALLS data for CD163 in the presence of Hp(1-1)Hb, with the left y-axis showing the differential refractive index, while the right y-axis shows the molecular weight. This is representative of two repeats for measurements at pH 7.5 with $Ca^{2+}$ and three repeats for all other buffers. The right-hand panel shows the binding of CD163, coated on the surface of an SPR chip, to Hp(1-1)Hb at 10 nM concentration. This is representative of three repeats. In both cases, the key above the graphs indicates the pH and the presence or absence of divalent cations. **c** Measurement of the uptake of fluorescently labelled Hp(2-2)Hb into HEK293 cells expressing CD163 in the absence (red line) and presence (orange line) of EGTA. The blue line shows non-receptor-mediated uptake into cells not expressing the receptor. Each point represents the mean of three replicates, and the error bars depict standard deviations.

measure the internalisation of fluorescently labelled Hp(2-2)Hb and to assess how effectively it competes for the uptake of non-fluorescent ligands (Fig. 5e). Fluorescent Hp(2-2)Hb was taken up into cells expressing monomeric CD163 with a similar concentration dependence as into cells which express wild-type CD163 (Supplementary Fig. 9a). However, while wild-type CD163 showed equivalent uptake of all HpHb variants, and weaker uptake of Hb (Fig. 3f), monomeric CD163 showed uptake of HpHb which varied with the number of HpSPHb heads, with most effective uptake of Hp(2-2)Hb, then Hp(1-1)Hb and finally HpSPHb, with $IC_{50}$ values of 27 nM, 0.14 µM and 0.64 µM (Fig. 5e). Monomeric CD163 also showed decreased uptake of Hb, with an $IC_{50}$ larger than 4 µM. Therefore, while the uptake of high-avidity Hp(2-2)Hb can be mediated by monomeric CD163, multimer formation allows the efficient uptake of lower-affinity or -avidity ligands, making CD163 a more promiscuous scavenger receptor.

## Discussion

CD163 is the archetypal example of a type I scavenger receptor, with a primary known role in the detoxification of Hb released by erythrocyte damage[12]. To achieve this, it internalises the 'essentially irreversible' complex[6,26] between cell-free Hb and serum Hp into macrophages. While free Hb rapidly forms complexes with Hp under healthy conditions, infectious or genetic diseases which increase erythrocyte lysis are associated with Hp depletion[4,5]. In these circumstances, it is beneficial for CD163 to mediate free Hb uptake. In contrast, uptake of Hp is not beneficial, as it would deplete Hp from the plasma without contributing to Hb detoxification. How does CD163 selectively recognise different HpHb isoforms, as well as Hb, without recognising

Hp? The findings presented here allow us to propose a model for how CD163 mediates specific binding and release of diverse ligands (Fig. 5f).

Our structural studies of CD163 provide insight into how it mediates ligand specificity, allowing uptake of ligands with diverse stoichiometries and architectures. Two or three CD163 protomers form an assembly, with the higher local concentrations on the membrane surface most likely favouring trimers. Each protomer presents an arm which provides potential ligand-contact surfaces, and these arms come together in different conformations to create a ligand-binding site for a single HpHb head. This ensures that HpHb variants with different numbers of heads are all endocytosed equally. The interaction between HpHb and CD163 involves eight SRCR domains, which mould around the ligand. This formation of a binding site by bringing together multiple small binding sites will allow promiscuity, with different combinations of these small surfaces creating binding sites for different ligands. Together with flexibility in the trimeric base, this ensures that binding sites can mould to accommodate different ligands and allows a trimeric receptor to recognise asymmetric binding partners. Most of the interactions form between CD163 and the Hb dimer, with fewer interactions with Hp. This leads to the uptake of free Hb but not unnecessary depletion of free Hp. The CD163 arms can also interact with one another, using the same surfaces which form ligand-binding sites. These arm-arm interactions will compete with arm-ligand interactions, and it is likely that this underpins a mechanism of autoinhibition, with weaker ligands being unable to compete with arm-arm interactions for receptor binding, therefore dampening the extent of CD163 promiscuity.

The structure also reveals the role of calcium ions, which mediate two distinct functions. Calcium ions are found at the interfaces between protomers in the base of CD163, mediating multimer formation. This allows the uptake of lower-avidity ligands and scavenger receptor promiscuity. The predominance of ion-mediated interactions at these interfaces also gives the interface rotational flexibility, allowing protomers to adopt different relative conformations as the receptor moulds around a ligand. In addition, calcium ions are found within the binding sites between arms and ligands, and this most likely makes each of these small interfaces more promiscuous, with stronger electrostatic binding rather than more extensive shape complementarity driving the interaction. Finally, the use of calcium in critical interfaces, together with pH, is important for ligand release. At the calcium concentrations in sera, CD163 will multimerise and bind ligands, while in low calcium, lower pH conditions in the endosomes, ligands will be released for degradation.

While this study was under submission, another study was also published, which presented structural insight into CD163 and its binding to HpHb[27]. These two studies share some similar findings. Firstly, both reveal the same overall architecture of the HpHb-bound CD163 dimer and trimer. Etzerodt et al. present 4.5 and 5.2 Å maps with docked AlphaFold models[27], which largely align with the higher resolution liganded dimer and liganded trimer structures presented here. Both studies also reveal the importance of calcium-dependent multimerisation and show calcium ions at the CD163-HpHb binding sites to be important for ligand uptake. Here, we generated more accurate molecular models of CD163 and its interaction with HpHb, as well as revealing the locations of calcium ions in the receptor base. We also present two additional conclusions. Firstly, our studies of the uptake of HpHb complexes constructed from different Hp isoforms provide deeper insight into how the architecture of the receptor allows it to bind promiscuously to different ligands. Secondly, our structures of unliganded CD163 reveal how arm-arm interactions can allow a mechanism of autoinhibition. Together, these two studies provide new insight into how CD163 functions, showing that it is built for promiscuous ligand binding and release and revealing how it mediates the essential physiological process of Hb detoxification.

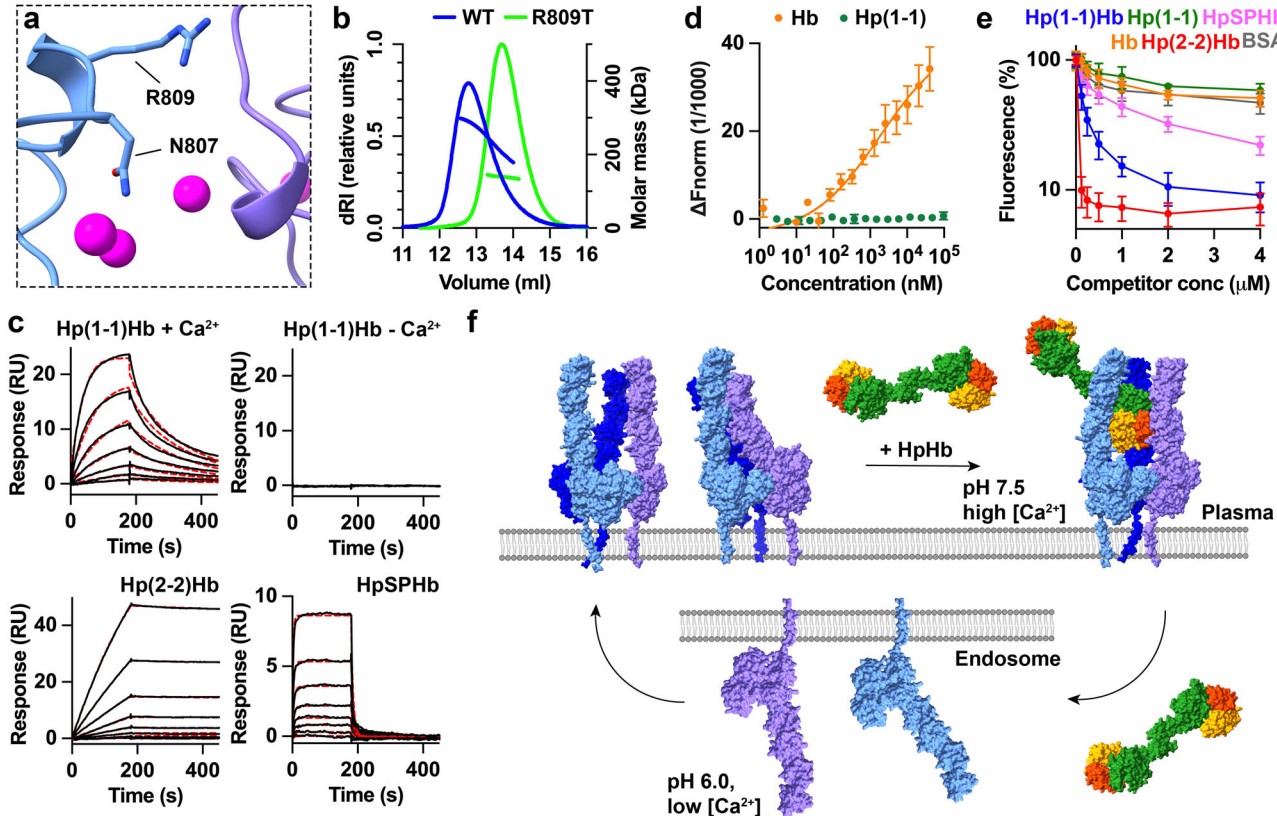

**Fig. 5 | Multimerisation of CD163 allows uptake of lower-avidity ligands. a** A close-up of the interface between $CD163_A$ and $CD163_B$, showing the location of N807, which becomes modified with an N-linked glycan in the R809T mutant. **b** SEC-MALLS analysis of wild-type (WT) CD163 ectodomain (blue) and the ecto-domain of the R809T mutant (green) with the left y-axis showing the differential refractive index, while the right y-axis displays the molecular weight. This is representative of two repeats at a concentration of 24 µM. **c** Assessment of the binding of CD163-R809T ectodomain, immobilised through a C-terminal biotin, to Hp(1-1)Hb, Hp(2-2)Hb and HpSPHb by SPR analysis. In the presence of $Ca^{2+}$, the sensorgrams represent twofold dilution series from a maximum concentration of 5 nM for Hp(1-1)Hb, 2.5 nM for Hp(2-2)Hb and 80 nM for HpSPHb. In the absence of $Ca^{2+}$, one injection of Hp(1-1)Hb at 10 nM is displayed. Data were shown as black lines, while fitting to a one-to-one binding model is depicted as dashed red lines. These are representative of $n = 3$. **d** Assessment of the binding of Hp (green) and Hb

(orange) to CD163-R809T ectodomain using MST. Each point represents the mean of three replicates, and the error bars are the standard deviation. **e** Measurement of the ability of different ligands to compete for the uptake of fluorescently labelled Hp(2-2)Hb into HEK293 cells transfected with CD163-R809T. BSA is included as a control for CD163-independent effects. Each point depicts the mean of three replicates, and the error bars are standard deviations. **f** A model for the CD163-mediated uptake of HpHb. At the cell surface, calcium mediates multimerisation of CD163. Two states exist in equilibrium, with the arms either free to bind to ligand or forming arm-arm interactions which occlude ligand-binding sites. On addition of ligand, the free arms of CD163 mould around the ligand to form a complex. On internalisation into the endosome, a drop in calcium concentration and pH results in monomerisation of CD163 and ligand release. CD163 can then be recycled back to the cell surface while HpHb is degraded and detoxified.

## Methods

### Expression and purification

The ectodomain of human CD163 (Uniprot Q86VB7-1 residues 46–1050) was cloned into pHLsec[28] and expressed with a C-terminal GAA-linker and a C-tag using the Expi293 Expression System (Thermo Fisher cat: A14635) according to the manu-facturer's instructions. Cell supernatant was harvested 6 days after transfection, passed through a 0.45-µm filter and applied to CaptureSelect C-tagXL Affinity Matrix (Thermo Scientific). Sub-sequently, the resin was washed with 30 column volumes (CV) of 20 mM Tris pH 7.5, 150 mM NaCl, and CD163 was eluted using 5 CV of 20 mM Tris pH 7.5, 2 M $MgCl_2$. The protein was further purified on a Superose 6 10/300 column (Cytiva) into 20 mM HEPES pH 7.5, 150 mM NaCl, 2.5 mM $CaCl_2$, flash frozen in liquid nitrogen and stored at – 80 °C. CD163-BAP was generated by fusing CD163 to a BAP tag, a GAA linker and the C-tag, and was expressed in the presence of 0.1 mM biotin. Site-directed muta-genesis of CD163-BAP was employed to generate the monomeric R809T variant, with mutations incorporated into the primers used for Gibson assembly. Both were purified as described for CD163.

### Production of HpHb complexes

Hb was obtained by injecting the supernatant of freshly lysed human erythrocytes on a Superdex 75 16/600 column (Cytiva), which was run using 20 mM HEPES pH 7.5, 150 mM NaCl, 2.5 mM $CaCl_2$. Hp(1-1) and Hp(2-2) extracted from human plasma were purchased from Sigma-Aldrich (SRP6507 and SRP6508). HpSP (Uniprot P00738-2 residues 89–347) and HprSP[19] (Uniprot P00739-1 residues 90–348) were cloned into pHLsec with a GAA linker and the C-tag, expressed in Expi293F cells for 6 and 5 days post-transfection, respectively, and C-tag purified as for the CD163 ectodomain. HpSP and HprSP were further purified on a Superdex 200 10/300 column (Cytiva) equilibrated in 20 mM HEPES pH 7.5, 150 mM NaCl, 2.5 mM $CaCl_2$.

Complexes of various Hp phenotypes with Hb were formed by mixing Hp with a molar excess of Hb. Hp(1-1)Hb and Hp(2-2)Hb were directly injected on a Superdex 200 10/300 and a Superose 6 10/300 column, respectively, and eluted into 20 mM HEPES pH 7.5, 150 mM NaCl, 2.5 mM $CaCl_2$. HpSPHb and HprSPHb were first subjected to C-tag purification before application on a Superdex 200 10/300 column.

The concentrations of HpHb complexes provided here referred to individual heads (HpSP plus an Hb dimer) rather than entire HpHb

molecules. Hb and Hp(1-1) concentrations were given for tetramers and dimers, respectively.

## Cryo-EM sample preparation and data collection

To prepare samples for cryogenic electron microscopy, 190 μg CD163 was mixed with 330 μg Hp(1-1)Hb in a total volume of 100 μl, injected on a Superose 6 3.2/300 column (Cytiva) and eluted into HBS with calcium. The peak fractions containing CD163:Hp(1-1)Hb were pooled and concentrated to 1.3 mg/ml. Quantifoil R 1.2/1.3 Cu 300 grids were glow-discharged for 40 s at 15 mA and mounted in a Vitrobot Mark IV (Thermo Scientific) operated at 4 °C and 100% humidity. Concentrated complexes were applied onto the grids, which were blotted for 2.5–4 s and plunge-frozen in liquid ethane. The grids were imaged on a Titan Krios G3 microscope (Thermo Scientific) equipped with a 300 kV emission gun, a K3 direct electron detector (Gatan) and a BioQuantum imaging filter (Gatan) with a slit width of 20 eV. EPU (Thermo Scientific) was used for automated data acquisition and was set to faster acquisition. Micrographs were collected with a defocus of −2.0 to −0.4 μm, a magnification of 58,149x, a pixel size of 0.832 Å and a dose of 41.68 e⁻/Å² over a period of 2.5 s. Three grids were subjected to data acquisition, yielding 39,445 movies (8303, 17,714 and 13,428, respectively).

Unliganded CD163 was concentrated to 1.3 mg/ml after gel filtration, and corresponding samples were prepared under the same conditions as CD163:Hp(1-1)Hb. Data collection was similar to that for the liganded complex, with the defocus set to −1.8 to −0.6 μm and a dose of 42.82 e⁻/Å² over 2.6 s. 19,412 micrographs were collected from a single grid.

## Cryo-EM image processing

Micrographs from the CD163:Hp(1-1)Hb dataset were pre-processed in SIMPLE 3.0[20]. Following motion correction and contrast transfer function (CTF) parameters estimation, micrographs were picked using a template from a previous test data collection. After 2D classification, 1,573,382 particles were extracted and exported into CryoSPARC[21], where the particles were further classified and used for 3D reconstruction. The resulting volume was used to re-pick the full dataset, resulting in 15,666,525 particles with a box size of 512 pixels, Fourier-cropped to 384 pixels.

Over the course of three rounds of 2D classification, classes containing poor particles were excluded, leaving 3,502,829 particles that were used for 3D reconstruction into seven classes. Classes for dimeric and trimeric CD163:Hp(1-1)Hb complexes containing most features underwent 3D classification via ab initio reconstruction and heterogeneous refinement, after which particles with nearly identical features were combined. 463,222 particles corresponding to the trimeric complex were re-extracted to the original pixel spacing of 0.832 Å and subjected to non-uniform refinement with optimisation of CTF parameters. This led to a volume at 3.2 Å resolution according to the gold-standard Fourier shell correlation (GSFSC) definition at 0.143. For the dimeric complex, 456,124 particles were re-extracted to remove the Fourier-crop and processed in a final non-uniform refinement step with per-particle defocus and CTF parameters optimisation, yielding a map at 2.8 Å resolution with the original pixel spacing of 0.832 Å.

Unliganded CD163 was processed similarly to CD163:Hp(1-1)Hb. After pre-processing and initial 2D classification in SIMPLE, 798,259 particles were exported to CryoSPARC. These particles were used to reconstruct a 3D template used to re-pick all micrographs. 13,772,487 particles were extracted using a box size of 512 pixels, down-sampled to 384 pixels and sorted in two rounds of 2D classification. This resulted in 3,308,159 particles that were used to generate seven ab initio volumes. A high-resolution dimer class, a trimer class with disordered arms and a trimer class with arm-arm contacts were identified. Two rounds of 3D classification were performed on particles corresponding to the dimer class, while particles corresponding to the trimer classes were 3D-classified once, after which particles with similar

features were combined. 723,620 particles corresponding to the dimer class were re-extracted to remove down-sampling and subjected to non-uniform refinement, yielding a volume at 3.1 Å resolution with a pixel size of 0.832 Å. 81,871 particles corresponding to the trimer class with disordered arms and 432,423 particles representing the trimer with arm-arm contacts underwent non-uniform refinement. This resulted in maps at 3.8 and 3.4 Å, respectively. All maps were sharpened by DeepEMhancer[29] using the default settings and were depicted using ChimeraX v1.7[30]. Local resolutions were estimated in cryoSPARC using unsharpened maps and depicted in ChimeraX using DeepEMhancer-sharpened maps.

## Model building and refinement

Trimeric CD163:Hp(1-1)Hb exhibited a higher resolution for the CD163 base, while subunit A of dimeric CD163:Hp(1-1)Hb most clearly resolved an arm of CD163. Therefore, SRCR domains 5–9 of the liganded complexes were built using ChimeraX by rigid-body fitting individual AlphaFold2[31]-predicted domains into the corresponding densities of trimeric CD163:Hp(1-1)Hb. Domains 2–4 were docked into the densities of the liganded dimer. Falsely built regions were corrected in Coot 0.9.8.8[32] before docking the corrected SRCR domain models into unmodelled densities of the liganded CD163 maps. These models were then manually readjusted in Coot.

A crystal structure of HpSPHb[19] (PDB: 4X0L) and Hp(1-1)Hb from a crystal structure of its complex with the *T. brucei* HpHb receptor (PDB: 4WJG) were docked into ligand densities of trimeric and dimeric CD163:Hp(1-1)Hb, respectively. Both models were initially subjected to optimisation using ISOLDE[33] and subsequent iterative real-space refinement in PHENIX[34] and manual correction in Coot.

To build the unliganded CD163 dimer, SRCR domains from the liganded dimer were individually rigid-body fitted into corresponding densities using ChimeraX[30] and manually adjusted in Coot. The unliganded dimer was refined as the liganded complexes. PDBePISA[35] was used to calculate ligand-binding interfaces and interfaces of arm-arm contact. Models were displayed using ChimeraX.

## Surface plasmon resonance

SPR was performed on a Biacore T200 SPR system (Cytiva) operated at 25 °C using the Biotin CAPture Kit, Series S (Cytiva). SPR buffer was either HBST (20 mM HEPES pH 7.5, 150 mM NaCl, 0.005% Tween-20) or MBST (20 mM MES pH 6.0, 150 mM NaCl, 0.005% Tween-20) supplemented with either 2.5 mM CaCl₂ or 2.5 mM MgCl₂ or neither. Ligand and analyte were buffer exchanged using 0.5 ml Zeba Spin Desalting columns (7 K MWCO, Thermo Scientific) into SPR buffer prior to experimentation.

Biotinylated CD163-WT-BAP or CD163-R809T-BAP diluted to 8 μg/ml were captured at 8 μl/min for 160 s, giving ~300 and 150 RU, respectively, with low coupling levels used to reduce surface density and reduce the likelihood of multiple monomeric CD163-R809T-BAP coupling sufficiently closely to simultaneously bind to a single ligand. Under binding conditions for CD163-WT-BAP, twofold dilution series of 10-0.078 nM Hp(1-1)Hb, 10-0.078 nM Hp(2-2)Hb, 20-0.16 nM HpSPHb or 20 nM HprSPHb were flowed through the chip at 30 μl/min for 3 min, followed by dissociation for 5 min. For CD163-R809T-BAP, twofold dilution series of 5-0.078 nM Hp(1-1)Hb, 2.5-0.019 nM Hp(2-2) Hb and 80-0.63 nM HpSPHb were used. Under non-binding conditions, 10 nM Hp(1-1)Hb and 20 nM HprSPHb were injected onto the chip. The chip surface was regenerated by injecting 6 M guanidine hydrochloride, 0.25 M NaOH at 10 μl/min for 80 s. Under binding conditions, experiments were conducted three times with three independent dilution series. Under non-binding conditions, a single concentration corresponding to the highest concentration observed under binding conditions was injected three times. Kinetic analysis was performed using BIAevaluation software v1.0 (Cytiva). Curves were depicted using GraphPad Prism v10.3.1.

## Fluorescent labelling of CD163 ligands

For MST, Hb and Hp(1-1) were fluorescently labelled using Alexa Fluor 647 NHS Ester (Thermo Scientific) according to the manufacturer's instructions. For uptake experiments, Hp(2-2) was labelled using Alexa Fluor 594 NHS Ester (Thermo Scientific). Unbound dye was removed by applying the sample on a Zeba Spin Desalting column and a Superdex 200 10/300 column and eluting it in HBS with calcium. Fluorescent Hp(1-1)Hb and Hp(2-2)Hb were produced by mixing fluorescent Hp(1-1) and Hp(2-2), respectively, with an excess of Hb and subsequent gel filtration.

## Microscale thermophoresis

MST was carried out on a Monolith NT.115 (NanoTemper) operated at 25 °C. Four different sets of binding partners were prepared in 20 mM HEPES pH 7.5, 150 mM NaCl, 2.5 mM $CaCl_2$, incubated for an hour at room temperature and loaded into Monolith Capillaries (NanoTemper) for measurement. In the first set, 200 nM fluorescent Hb was mixed with a twofold dilution series of CD163-WT (24.4 µM–0.74 nM), which was measured at 10% LED and 20% MST power. In the second set, 500 nM Hp(1-1) was incubated with CD163-WT (37.9 µM–1.2 nM) and was recorded at 8% LED and 20% MST power. In the third set, 200 nM Hb was mixed with CD163-R809T-BAP at 41.5 µM–1.3 nM and was measured at 10% LED and 20% MST power. In the fourth set, 1.0 µM Hp(1-1) was mixed with 96.3 µM–2.9 nM CD163-R809T-BAP and was analysed at 6% LED and 20% MST power. Every interaction was characterised in technical triplicate. Raw data were extracted from NT Analysis v1.5 (NanoTemper), and a fluorescence baseline offset was applied to individual replicates before they were combined and fitted using the 'Sigmoidal, 4PL, X is concentration' model in GraphPad Prism.

## Size-exclusion chromatography coupled with multi-angle laser light scattering

SEC-MALLS was conducted at room temperature on a Shimadzu Prominence UFLC System connected to a Superose 6 10/300 column, a Dawn Helios 8+ light scattering detector (Wyatt Technology) and an Optilab T-rEX refractive index detector (Wyatt Technology). To test the influence of the CD163 concentration on its multimeric state, duplicates of 100 µl samples containing monomer concentrations of 3.47, 6.11, 11.9 or 24.2 µM CD163-WT were measured in HBS (20 mM HEPES pH 7.5, 150 mM NaCl) with 2.5 mM calcium. Similarly, duplicates with monomer concentrations of 2.28, 3.47, 4.65 or 6.93 µM CD163-WT and 1.18 µM Hp(1-1)Hb were analysed. To probe the role of calcium and pH, triplicates containing 3.47 µM CD163-WT and 1.18 µM Hp(1-1)Hb were analysed in HBS, HBS with 2.5 mM $MgCl_2$, MBS (20 mM MES pH 6.0, 150 mM NaCl) or MBS with 2.5 mM $CaCl_2$. To characterise the effect of R809T on the multimeric state of CD163, duplicates containing 3.41 or 24.2 µM CD163-R809T-BAP were measured in HBS with calcium. Data were analysed using ASTRA v6.1 (Wyatt Technology) and plotted in GraphPad Prism.

## Generation of stably transfected, CD163-expressing HEK293 cells

The Flp-In-293 system (Thermo Fisher cat: R78007) was used to generate stably transfected, CD163-expressing cells according to the manufacturer's instructions. The secretion signal sequence of pHLsec, full-length human CD163 (Uniprot Q86VB7-1 residues 46–1156, wildtype or R809T), a GSG linker and GFP (Uniprot P42212 residues 2–238) were cloned into pcDNA 5/FRT (Invitrogen). This expression vector and pOG44 were used to transfect ~600,000 Flp-In-293 T-REx cells (Thermo Scientific) at a 1:9 ratio using TransIT-LT1 Transfection Reagent (Mirus Bio) according to the manufacturer's instructions. Cells were passaged in Dulbecco's Modified Eagle Medium (DMEM, Gibco) with 10% foetal calf serum (FCS, Gibco) a day after transfection and selected in DMEM with 10% FCS, 15 µg/ml blasticidin S (Gibco) and

100 µg/ml hygromycin B (VWR) 2 days post-transfection. After selection, cells were sorted using FACSAria Fusion (BD Biosciences) to obtain populations with high, consistent and equivalent CD163 expression for wildtype and mutant receptors.

## Ligand uptake assay

For Hp(2-2)Hb uptake in the absence of competitors, -200,000 CD163-WT- or CD163-R809T-expressing cells or untransfected cells were washed in PBS. After incubation in DMEM with 25 mM HEPES, 0–600 nM Alexa Fluor 594-labelled Hp(2-2)Hb and optionally 2 mM EGTA for 30 min at 37 °C, 5% $CO_2$, cells were washed in PBS, trypsinised and washed again. Following the addition of DRAQ7 (Abcam) to 3 µM, cells were measured on a LSRFortessa X-20 Cell Analyzer (BD Biosciences), equipped with a 488 and 561 nm laser to detect CD163-GFP and labelled Hp(2-2)Hb, respectively. For uptake of Alexa Fluor 594-labelled transferrin (Thermo Scientific, T13343), untransfected cells were handled as described for Hp(2-2)Hb uptake.

For uptake in the presence of competitors, CD163-WT- or CD163-R809T-expressing cells were incubated with 50 nM Alexa Fluor 594-labelled Hp(2-2)Hb and 0–4 µM unlabelled ligand and were processed as described above.

All uptake experiments were performed in triplicate, with cells for each sample incubated with fresh ligand and measured on three different days. To quantify uptake, the mean fluorescence of all live, single cells was calculated using FlowJo v10.10 and plotted in GraphPad Prism. In the uptake series performed with competitors, slight batch-to-batch differences were observed in the absolute fluorescence signal for the sample at 0 nM competitor. Therefore, fluorescence values in these series were normalised by defining the fluorescence at 0 nM competitor as 100%. In competition experiments, $IC_{50}$ values were defined as the concentration at which the fluorescence reaches 50%.

## Reporting summary

Further information on research design is available in the Nature Portfolio Reporting Summary linked to this article.

## Data availability

Cryo-EM maps are deposited in the Electron Microscopy Data Bank under accession codes EMD-52078 for dimeric CD163:HpHb, EMD-52079 for trimeric CD163:HpHb, EMD-52080 for unliganded dimeric CD163 with arm-arm contacts, EMD-52081 for unliganded trimeric CD163 with arm-arm contacts and EMD-52082 for unliganded trimeric CD163 with disordered arms. Atomic coordinates are deposited in the Protein Data Bank under accession codes 9HEJ for the liganded CD163 dimer, 9HEK for the liganded trimer and 9HEL for the unliganded dimer. Source data are provided with this study. Source data are provided with this paper.

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

## Acknowledgements

This work was funded by the Wellcome Trust. M.K.H. is a Wellcome Investigator (220797/Z/20/Z) and R.X.Z. was funded by the graduate programme in Cellular Structural Biology (218482/Z/19/Z), Magdalen College and the Clarendon Fund. Cryo-EM data were collected at the COSMIC facility, and we thank Rishi Matadeen, Teige Matthews-Parmer, Joe Caesar and Ed Lowe for support with data collection and processing. We acknowledge Hannah Ivison for lab management. We are grateful to David Staunton for help with biophysical methods and to Robert Hedley and Vasiliki Tsioligka for assistance with flow sorting. We thank Ian Gibbs-Seymour for advice on the generation of stable cell lines and Thomas Carroll for advice on uptake experiments. We thank Dara Thaker for technical assistance.

## Author contributions

R.X.Z. conducted all experimental work. R.X.Z. and M.K.H. conceived and planned the study, designed experiments, analysed the data and wrote the manuscript.

## Competing interests

The authors declare no competing interests.
