## [Transparent Peer Review file · Nature Communications]

Scavenger receptor CD163 multimerises to allow uptake of diverse ligands

Corresponding Author: Professor Matthew Higgins

Version 0:

Reviewer comments:

Reviewer #1

(Remarks to the Author)

The manuscript by Zhou and Higgins entitled "Scavenger receptor CD163 multimerises to allow uptake of diverse ligands" presents a large amount of significant data, that however, still requires more extensive description, in-depth interpretation and comparison with literature. Importantly, a paper on the same topic has been recently published (Etzerodt, A. et al. The Cryo-EM structure of human CD163 bound to haptoglobin-hemoglobin reveals molecular mechanisms of hemoglobin scavenging. doi.org/10.1038/s41467-024-55171-4), therefore the authors should consider critically highlighting the novelty of their study with respect to it.

I recommend the authors to undertake an extensive revision of the work starting from the main points outlined below and I will postpone a more detailed revision for further review, if needed.

1. The abstract sounds very descriptive, with few references to specific data and or numeric data. This results in a text which does not convey the main findings of the work. Furthermore, references should be removed from abstract.
2. Introduction is very short. More insight on relevant literature about CD163 structure, role of calcium, differences, if any, in the internalization efficiency of Hp2 and Hp1, etc...is mandatory.
3. Conclusions should be discussed considering existing literature
4. The paper contains very important findings that are underrated and not sufficiently highlighted (eg. Line 80, 99-104, and 158-159)
5. CryoEM data:
 - a. The resolution would allow more details on the specific interactions between CD163 and ligands. Furthermore, the pdb model should consider all calcium-mediated interactions. A table summarizing the PISA-estimated contacts would be useful. Are PTMs visible? Their role in the interaction could be investigated.
 - b. The authors do not provide an explanation for why the second protomer of HpHb is not visible.
 - c. Could structural analysis suggest a model for internalization triggering? It is important to note that R809T is also internalized, so oligomerization cannot be the determining factor.
 - d. Greater clarity regarding the structural analysis of the dimer and trimer is necessary. Sometimes, information coming from different models seem to be mixed up (Fig 2d).
6. Overall, the work requires more detailed comments on data and more thorough data analysis. Most of the results are reported without any background: why was Hp1 instead of Hp2 chosen? How was the stoichiometry for cryo-EM specimens preparation chosen? (molar excess of Hp1Hb might impact on the assembly that is observed on the grids?) why is calcium used in specimens' preparation? Why was HpSP used? Why was Hp2-2 used competition assays instead of Hp1-1?
7. Autoinhibition mechanism: is it fully supported by data? E.g. is the number of interactions with ligands higher than the intrasteric ones? Could the structure be an artifact of sample preparation/cryo-EM conditions? Are there similar examples in literature?
8. The section discussing the monomeric variant is particularly challenging for the general reader, as the rationale behind it is only briefly explained. How do the authors demonstrate the formation of N-glycosylation in R809T? In addition to providing further insights into the role of oligomerization, the R809T model could be exploited to determine which species are present in various experiments (e.g., in SPR with WT, is oligomerization expected? how do the authors distinguish the monomer from other oligomers?).
9. Role of calcium is introduced abruptly w/o prior explanation of its potential role neither in the introduction nor in the results. The chosen calcium concentration is 2.5 mM, why?

10. The authors should provide more in-depth comparison between ligated and unligated structures.
11. I suggest using molarity instead of mg/mL to improve readability

Reviewer #2

(Remarks to the Author)

Overview and significance

The manuscript by Richard X. Zhou presents a thorough structural and functional investigation of CD163, the macrophage scavenger receptor for hemoglobin clearance. Using a combination of high-resolution cryo-electron microscopy (down to 2.8–3.2 Å in certain classes), SEC-MALLS, SPR, and cell-based uptake assays, the authors provide novel insights into how CD163 flexibly assembles as dimers and trimers to bind multiple hemoglobin-containing ligands with varying stoichiometries. Significant is the demonstration of (1) High-resolution detail of the receptor “base” and “arms” in both ligand-bound and unliganded states. (2) A previously undescribed autoinhibition mechanism, wherein the receptor arms form interprotomer contacts that occlude ligand-binding surfaces without ligand. (3) The role of calcium in assembling the multimeric receptor base and stabilizing multiple small interfaces around the hemoglobin–haptoglobin (HpHb) ligand. (4) How CD163 flexibility and high local avidity allow the uptake of diverse HpHb complexes and free hemoglobin (at higher concentrations) and how it discriminates against free haptoglobin alone.

The data support a model in which CD163 multimerizes at physiological calcium levels, enabling robust ligand capture. At the same time, the acidic, calcium-depleted endosomal environment triggers both ligand release and receptor disassembly. These findings advance the mechanistic understanding of hemoglobin detoxification.

Main results and methods

Cryo-EM: Achieved resolutions of ~2.8 Å for dimeric CD163-HpHb, ~3.2 Å for trimeric complexes, and ~3.1–3.4 Å for unliganded forms.

Biophysical Assays (SEC-MALLS, SPR, MST): Quantified how calcium drives oligomerization and measured nanomolar to micromolar affinities for multiple ligands, including Hp(1-1)Hb, Hp(2-2)Hb, truncated HpSPHb, free Hb, and free Hp.

Cell-based Uptake: Demonstrated that forced monomeric CD163 is substantially less efficient at internalizing lower-affinity complexes, underscoring the functional necessity of receptor oligomerization.

A central issue that has to be considered by the authors and editors is the recently published paper by Etzerodt et al., “The Cryo-EM structure of human CD163 bound to haptoglobin-hemoglobin reveals molecular mechanisms of hemoglobin scavenging”, also in Nature Communications, two months ago. (<https://www.nature.com/articles/s41467-024-55171-4>)

Despite the broad overlap with Etzerodt’s paper, the Zhou manuscript presents additional progress in three main domains.

(1) Higher-Resolution Structural Insights: The authors deliver a deeper atomic-level description of CD163–HpHb interactions, including side-chain placements and a unique visualization of the unliganded receptor’s autoinhibitory arrangement. (2) Broader Ligand Range: By comparing HpHb from different isoforms, engineered Hp domains, and haptoglobin-related protein, this work elucidates the receptor’s flexibility and specificity more comprehensively than the Etzerodt paper. This is important, given that an in human ligand heterogeneity exists, provided by the genetic polymorphism in haptoglobin. (3) Novel Mechanistic Angle: The new autoinhibition data explains how CD163 restricts off-target uptake, extending the fundamental biology beyond what has been described.

Therefore, my primary suggestion is that the authors incorporate a clear comparison with Etzerodt’s paper, emphasizing unique data that strongly argue for the importance and novelty of this submission. The authors should highlight the distinctions in resolution, experimental scope, and mechanistic emphasis, ensuring readers understand how their paper complements and extends the published data. The two papers provide a very complete structural explanation of hemoglobin clearance via CD163.

Reviewer #3

(Remarks to the Author)

Version 1:

Reviewer comments:

Reviewer #1

(Remarks to the Author)

The revised version of the manuscript by Zhou and Higgings entitled “Scavenger receptor CD163 multimerises to allow

uptake of diverse ligands" shows substantial improvements compared to the original submission. I am pleased to see that the authors have adapted the text to the longer format allowed by Nature Communications, as requested by the reviewer. The data presented are of very high quality, and the scientific question addressed is important. However, I still believe that—aside from personal preferences about how data should be presented and where to draw the line between interpretation and speculation—the manuscript requires further discussion/description on some findings. While the authors argue that it is not a reviewer's role to judge the presentation of data, as long as it is technically sound, I must respectfully disagree. High-quality data is the foundation, but it is indeed a reviewer's responsibility to assess whether it is also effectively presented.

As stated in the guidelines one of the factors to keep in mind is:

"The quality of the data — whether they are technically sound, obtained with appropriate techniques, analysed and interpreted carefully, and presented in sufficient detail."

I will provide more specific comments below, but I would like to first highlight two major points:

1) Abstract

I had previously suggested to enrich the abstract with more specific data. The authors have chosen to retain the original version, and while I will not insist further, I believe that explicitly mentioning that the investigation involves the structural characterization of two ligation states of CD163—highlighted as a novelty over the work by Etzerodt et al. in the conclusions by the authors—would be beneficial.

2) Conclusions

The conclusions mainly comment on the results with only limited discussion in light of existing literature. While I agree with the authors that the conclusion might focus on results, if the editorial office allows a manuscript to omit a separate "Discussion" section, then this is acceptable.

Major Points

1) Sample Preparation for Cryo-EM

While I would have approached grid preparation differently (e.g., by testing different stoichiometric ratios), I understand that revising the experimental setup retrospectively is not feasible. However, it should be clearly acknowledged that using varied stoichiometries could have helped resolve different assemblies and potentially led to a more complete understanding of the complex.

For the sake of reproducibility, the authors should specify not only that they used a molar excess of HpHb relative to CD163, but also the exact concentrations employed. It should be clarified if these concentrations differ from those used in SEC-MALLS, and therefore, the two experiments are not directly comparable.

Related to this, a thorough discussion of Extended Data Fig. 1c is warranted. Specifically, the authors should comment on the result that increasing the total protein concentration appears to reduce the relative abundance of high to low molecular weight species, which is not obvious considering that the stoichiometric ratio should not change. Finally, it would be highly beneficial if the authors could provide a supplementary table listing the molecular weights determined by SEC-MALLS alongside the theoretical molecular weights of the species they believe they have isolated.

2) SPR Analysis

Two major concerns need to be addressed:

1. The authors immobilize biotinylated CD163 and its R809T variant on SPR chips to measure ligand binding affinities. However, it is unclear what assembly (monomeric, dimeric, trimeric or a mixture) they assume for the immobilized CD163. While trimers are briefly mentioned (line 176), this is not experimentally demonstrated. While it is likely that R809T is immobilized as a monomer, the authors should provide evidence that CD163 forms trimers upon immobilization or explicitly state that the oligomeric state remains uncertain. This clarification is particularly important given that the K_d values determined here differ from those reported by Etzerodt et al., who used an engineered, stable trimeric form of CD163.
2. The K_d value reported for Hp(2-2)Hb shows an uncertainty equal to the value itself. The authors should explain how this level of uncertainty affects their conclusion—that monomeric CD163 binds Hp(2-2)Hb with similar or higher affinity than trimeric CD163—a statement that is currently difficult to interpret.

3) Monomeric CD163

Although the authors have made some minor changes to the narrative, the description of this mutant remains insufficient. While it is clear that the mutation introduces a glycosylation recognition site, the rationale behind its design is not. Where is the mutated residue located? Why was it chosen (e.g., based on previous structural or mutagenesis studies)? Given that this is a non-standard method for disrupting protomer-protomer interactions, the authors should clearly explain the rationale and any expected advantages.

In relation to the ligand uptake the authors should state if the two receptor forms (wt and R809T) are expressed to the same levels.

4) Role of Calcium

The calcium-binding sites are only superficially described. The authors primarily refer the reader to a supplemental table for residue identities and site positions. They should clearly indicate whether the identified sites are consistent with previously reported metal-binding motifs in SRCR family proteins and highlight any that are novel. A more detailed comment on the calcium-binding sites located at the base of the receptor would also be valuable—especially since these are mentioned in the conclusion in reference to the Etzerodt study.

Additionally, the calcium concentration used is described as "physiological", in both the rebuttal letter and the main text, however it should be acknowledged that this concentration has been previously shown (ref. 14) to be saturating.

Minor comments:

Line 46: In this context, "different ligands" should be revised to "different assemblies of the same ligand" or a similar phrase.

Line 52: Add reference 27.

Line 55: Free Hb in solution is expected to predominantly exist in the dimeric form, at least in the case of modest hemolysis.

Line 71: Cite Madsen (ref. 14) to support the choice of calcium concentration, or alternatively, cite other relevant literature.

Line 74: Add reference to Extended Data Fig. 1.

Line 104: The notation "CD163A" has not been defined previously. While its meaning is somewhat self-evident, a brief description would be beneficial.

Lines 176 and 215: The values provided refer to dissociation constants, not affinities.

Line 280: "Our structural studies of CD163 show how it mediates ligand specificity." The reviewer notes that the authors have not fully demonstrated ligand specificity at the structural level, as they have not determined the receptor's structures in complex with all relevant ligands. Rather, a combination of biochemical methods and selected structural data has been used to propose a hypothesis.

Lines 338-339: Please provide more details on the mutagenesis procedure employed.

Lines 355-357: Please clarify the meaning of "individual heads" in this context.

Legend to Extended Data Fig. 3: The same information about the blue and black parts, as presented in Extended Data Fig. 2, should be included.

In their rebuttal letter, the authors state: A structure with two trimeric receptors attached to the membrane, forming complexes with an HbHp dimer, is not physiologically relevant. However, prior data (Etzerodt, 2024) considered this possibility, and CD163 can also exist in a soluble form. The authors may wish to comment on this.

Reviewer #2

(Remarks to the Author)

The authors have addressed all my concerns. I want to congratulate the authors on this excellent work.

Reviewer #3

(Remarks to the Author)

Version 2:

Reviewer comments:

Reviewer #1

(Remarks to the Author)

The revised version of the manuscript by Zhou and Higgins, entitled "Scavenger receptor CD163 multimerises to allow uptake of diverse ligands", has been largely modified in line with the reviewers' suggestions. Therefore, it can be considered for publication.

There are, however, a few minor points that the authors may want to address in the final version:

1. Regarding the large error associated with the K_d values calculated for wild-type CD163 binding to Hp(2-2)Hb, I agree with the authors that the high-affinity interaction likely hinders accurate estimation of dissociation constants. However, the K_d value for R809T CD163 binding to Hp(2-2)Hb—which is even lower—has a comparatively smaller error. The authors should clarify in the text that the large error specific to this K_d measurement limits its utility for meaningful comparison with other data.

2. The authors note that the insertion of glycosylation sites is a strategy used by them and others to disrupt protein-protein interactions. Including a reference to a publication where this technique has been successfully applied would be highly beneficial to readers.

3. In line 71, the authors cite Madsen et al., but do not explicitly state that the calcium concentration used in that study was confirmed to be saturating. I suggest adding this detail to clarify the experimental context.

Reviewer #3

(Remarks to the Author)

REVIEWER COMMENTS

Reviewer #1 (Remarks to the Author):

The manuscript by Zhou and Higgings entitled “Scavenger receptor CD163 multimerises to allow uptake of diverse ligands” presents a large amount of significant data, that however, still requires more extensive description, in-depth interpretation and comparison with literature. Importantly, a paper on the same topic has been recently published (Etzerodt, A. et al. The Cryo-EM structure of human CD163 bound to haptoglobin-hemoglobin reveals molecular mechanisms of hemoglobin scavenging. doi.org/10.1038/s41467-024-55171-4), therefore the authors should consider critically highlighting the novelty of their study with respect to it.

We thank the reviewer for highlighting the large amount of significant data which we present in this manuscript.

This manuscript was originally submitted, on 15th November 2024 to another Nature journal which has a shorter format and was later redirected to Nature Communications. This is why the format is not the standard for Nature Communications. We have therefore reformatted the manuscript into the longer Nature Communications format for this revision.

Our work was submitted before the publication of Etzerodt and was therefore independent and not done in the context of Etzerodt. It is not a ‘follow-up’ study based on their ideas. It is therefore not appropriate for us to extensively rewrite our manuscript in the context of their work as we did not know their ideas before, or while we conducted our study. Instead, it is of more value for the community to see our independent ideas and independent findings. Indeed, it is the enlightened policy of Nature Communications to disregard ‘competing’ work published while a paper is under submission <https://doi.org/10.1038/s41467-020-17817-x>. We have therefore not extensively rewritten the manuscript in response to Etzerodt. However, we have put a section (lines 311-326) in the conclusion to make clear that their manuscript was published while ours was under submission and to highlight some similarities and differences to help readers to compare the two works.

I recommend the authors to undertake an extensive revision of the work starting from the main points outlined below and I will postpone a more detailed revision for further review, if needed.

We disagree with the reviewer that an ‘extensive’ revision is required. We also disagree with the reviewer’s view that multiple rounds of revision are appropriate. The purpose of reviewing is to highlight whether the data is robust, experiments are correctly done and that the conclusions drawn are supported by the data. The data was all presented in the first version of the paper and so any concerns about the data should have been highlighted in the first review. None were highlighted. It is not appropriate reviewing practice to ‘postpone a more detailed revision’ for a second round of revisions and so I presume that this is a misunderstanding and that the reviewer did not intend this.

1. The abstract sounds very descriptive, with few references to specific data and or numeric data. This results in a text which does not convey the main findings of the work. Furthermore, references should be removed from abstract.

We disagree that the abstract does not describe the data. Lines 7-18 outline the new data that we present in this manuscript, such that description of our new findings comprises more than half of the abstract. With the abstract size limit of 200 words it is not possible to include more information in the abstract. We have however removed the references to fit this to Nature Communications format.

2. Introduction is very short. More insight on relevant literature about CD163 structure, role of calcium, differences, if any, in the internalization efficiency of Hp2 and Hp1, etc...is mandatory.

We have now adapted the introduction to the expanded format of Nature Communications and added some paragraphs about what was known about CD163 structure and function, as well as the role of calcium in uptake (lines 38-56). This information was previously found in the appropriate location within the results sections. Previous papers did not show a robust and quantitative

comparison of uptake of Hp(1-1)Hb and Hp(2-2)Hb and this is an outcome of this paper rather than a finding from the literature.

3. Conclusions should be discussed considering existing literature

It is our view that this has been done. This is not an extensive literature review, but a novel research paper and our view is that we do give appropriate respect to the work of others, while focusing primarily on describing our novel findings.

4. The paper contains very important findings that are underrated and not sufficiently highlighted (eg. Line 80, 99-104, and 158-159).

We thank the reviewer for highlighting that these lines describe very important findings. We agree. However, our style of writing is not to overstate our findings but instead to present our findings in a concise and clear format. All the highlighted findings are clearly stated and are discussed in the conclusions, as well as being drawn into the model in Figure 5f. We think that this is clear and appropriate for scientific writing.

5. CryoEM data:

a. The resolution would allow more details on the specific interactions between CD163 and ligands. Furthermore, the pdb model should consider all calcium-mediated interactions. A table summarizing the PISA-estimated contacts would be useful. Are PTMs visible? Their role in the interaction could be investigated.

We thank the reviewer for the suggestion and have now included a new interaction table (Extended Data Table 3) which clearly shows the interactions made with liganded dimer and trimer and unliganded dimer. This will allow the reader to easily read across to see the degree to which interactions are conserved or variable in these different binding modes.

b. The authors do not provide an explanation for why the second protomer of HpHb is not visible.

We cannot be sure why the second protomer is not viable, although we can be sure that it is not. However, we have added some extra sentences in lines 79-85 to speculate about whether this is due to flexibility. What is clear from our findings is that this is not a physiologically relevant question as, with CD163 attached to the membrane and HpHb protruding from the CD163 trimers as shown in Figure 5, a second CD163 trimer couldn't readily bind to the other half of the dumbbell when membrane associated.

c. Could structural analysis suggest a model for internalization triggering? It is important to note that R809T is also internalized, so oligomerization cannot be the determining factor.

The literature suggests that CD163 internalisation is constitutive and not triggered by ligand binding. However, this is not a question which we have investigated through experiments in this study and so we prefer not to speculate as it is outside the scope of our work.

d. Greater clarity regarding the structural analysis of the dimer and trimer is necessary. Sometimes, information coming from different models seem to be mixed up (Fig 2d).

In our view, the manuscript is clear in its description of the dimer and trimer. The data in Figure 2d is not 'mixed up', but the legend states clearly what the colour code means. The aim of this figure is to show the reader that the residues involved in binding to HpHb substantially overlap with the residues involved in arm-arm interactions. We are confident that the presentation is high quality and clear.

We have however added a table which shows all interactions between CD163 and HpHb in both dimeric and trimeric arrangements as well as arm-arm interactions involved in the auto-inhibited state. Hopefully this should provide the reader interested in specific detail with the opportunity to answer their questions. The PDBs and structural data will also be available for readers to explore.

6. Overall, the work requires more detailed comments on data and more thorough data analysis. Most of the results are reported without any background: why was Hp1 instead of Hp2 chosen? How was the stoichiometry for cryo-EM specimens preparation chosen? (molar excess of Hp1Hb might impact on the assembly that is observed on the grids?) why is calcium used in specimens' preparation? Why was HpSP used? Why was Hp2-2 used competition assays instead of Hp1-1?

With more words available to use now, we have been able to increase our description of why we took different decisions, in addition to stating what we did.

In the case of complex assembly, we describe lines 65-68 why we chose Hp1. In lines 71-73 we describe why we chose the stoichiometry of mixing, based on the knowledge at the time. Note that there is no need for concern about excess HpHb on the grids as the complex was separated from free HpHb by size exclusion chromatography (lines 73-74) and no free HpHb was seen.

We describe why we used 2.5mM CaCl₂ in line 70-71 and the introduction now describes previous data showing the importance of calcium (lines 40-45), to complement the existing discussion of calcium within the results and discussion sections.

HpSP was only used in uptake experiments and the rationale is provided in lines 159-161 and is also illustrated in Figure 3c.

We have now described why Hp(2-2)Hb was used in uptake assays in lines 185-188.

7. Autoinhibition mechanism: is it fully supported by data? E.g. is the number of interactions with ligands higher than the intrasteric ones? Could the structure be an artifact of sample preparation/cryo-EM conditions? Are there similar examples in literature?

We did attempt to devise an uptake experiment to test this autoinhibition mechanism, for example aimed to make a mutation which would disrupt the arm-arm interactions in the unliganded, auto-inhibited state, allowing us to determine whether disrupting these interactions would increase ligand uptake. However, this was not possible as the same residues which form these arm-arm interactions also interact with the ligand. We have therefore not been able to test our theory in an uptake study.

Nevertheless, the structural data is supportive of our proposal of an auto-inhibition mechanism.

Firstly, the clear overlap between the arm-arm interactions and arm-ligand interactions (Figure 2) mean that they are mutually inconsistent, meaning that, should arm-arm interactions occur on a macrophage surface, they will be in competition with ligand binding.

Several lines of evidence suggest that HpHb will be able to compete with the arm-arm interactions. First is that we do not see these interactions in our cryoEM studies in the presence of ligand. We have made this clearer with a new sentence in line 120-121. Secondly, we followed the reviewer's excellent suggestion and quantified the surface areas. This shows that the surface areas of the CD163-HpHb interfaces are larger than the surface area of the arm-arm interactions (lines 135-143) suggesting that HpHb will be able to outcompete the autoinhibited state.

Therefore while this is a hypothesis (made clear in lines 141-143), the structural data is supportive of this model.

Autoinhibition mechanisms have been seen in other cases, but these are very different receptors and so we do not think that it would be informative to compare in this case.

8. The section discussing the monomeric variant is particularly challenging for the general reader, as the rationale behind it is only briefly explained. How do the authors demonstrate the formation of N-glycosylation in R809T? In addition to providing further insights into the role of oligomerization, the R809T model could be exploited to determine which species are present in various experiments (e.g., in SPR with WT, is oligomerization expected? how do the authors distinguish the monomer from other oligomers?).

We have re-written the section about the introduction of the R809T mutation to attempt to make it clearer (lines 233-234). From our perspective it is clearly described.

The aim of the glycosylation mutant was to block multimerization. We have therefore not aimed to assess whether the glycan is present (although an increase in gel mobility suggests that it is) but instead show in Figure 5b that R809T is monomeric at concentrations at which the wild-type receptor is a multimer, while also binding to HbHb, suggesting functionality. The mutation has therefore achieved its objective.

We are not clear what experiments the reviewer is proposing here. We have conducted the experiments which we think are biologically relevant. Namely:

- (i) Demonstrating that calcium doesn't just affect multimerization, but also directly affects ligand binding, as it affects binding to the monomer mutant. This is lines 229-242, with a small addition for clarity.
- (ii) Showing whether the monomer mutant binds to a wide panel of ligands in the same affinity as multimeric CD163 (lines 243-253).
- (iii) Showing whether the monomer mutant affects uptake of the wide panel of ligands (lines 254-267)

In our view, this section of the paper is comprehensive, clear and contains all experiments required for us to draw our conclusions.

9. Role of calcium is introduced abruptly w/o prior explanation of its potential role neither in the introduction nor in the results. The chosen calcium concentration is 2.5 mM, why?

We have added mention of calcium to the introduction (lines 40-45) and make clear that the concentration used throughout the study was selected to match physiological serum calcium concentrations (line 71-72).

10. The authors should provide more in-depth comparison between ligated and unligated structures.

We have added a couple of extra sentences in lines 121-124 to highlight that the bases of the liganded and unliganded dimers are similar but the arms are different. Together with Figure 2b, which shows liganded dimer and trimer side-by-side, with an equivalent comparison for liganded and unliganded trimer in Extended Data Figure 5, this should allow the reader to clearly make this comparison.

11. I suggest using molarity instead of mg/mL to improve readability

We have changed to use of molarity for studies in which multimerization of CD163 and its complexes were studied. In describing sample preparation for cryoEM, we still use mg/ml as this is more standard.

Reviewer #2 (Remarks to the Author):

Overview and significance

The manuscript by Richard X. Zhou presents a thorough structural and functional investigation of CD163, the macrophage scavenger receptor for hemoglobin clearance. Using a combination of high-resolution cryo-electron microscopy (down to 2.8–3.2 Å in certain classes), SEC-MALLS, SPR, and cell-based uptake assays, the authors provide novel insights into how CD163 flexibly assembles as dimers and trimers to bind multiple hemoglobin-containing ligands with varying stoichiometries. Significant is the demonstration of (1) High-resolution detail of the receptor “base” and “arms” in both ligand-bound and unliganded states. (2) A previously undescribed autoinhibition mechanism, wherein the receptor arms form inter-protomer contacts that occlude ligand-binding surfaces without ligand. (3) The role of calcium in assembling the multimeric receptor base and stabilizing multiple small interfaces around the hemoglobin–haptoglobin (HpHb) ligand. (4) How CD163 flexibility and high

local avidity allow the uptake of diverse HpHb complexes and free hemoglobin (at higher concentrations) and how it discriminates against free haptoglobin alone.

The data support a model in which CD163 multimerizes at physiological calcium levels, enabling robust ligand capture. At the same time, the acidic, calcium-depleted endosomal environment triggers both ligand release and receptor disassembly. These findings advance the mechanistic understanding of hemoglobin detoxification.

Main results and methods

Cryo-EM: Achieved resolutions of ~ 2.8 Å for dimeric CD163-HpHb, ~ 3.2 Å for trimeric complexes, and ~ 3.1 – 3.4 Å for unliganded forms.

Biophysical Assays (SEC-MALLS, SPR, MST): Quantified how calcium drives oligomerization and measured nanomolar to micromolar affinities for multiple ligands, including Hp(1-1)Hb, Hp(2-2)Hb, truncated HpSPHb, free Hb, and free Hp.

Cell-based Uptake: Demonstrated that forced monomeric CD163 is substantially less efficient at internalizing lower-affinity complexes, underscoring the functional necessity of receptor oligomerization.

We thank the reviewer for noting and highlighting the advances presented in our manuscript.

A central issue that has to be considered by the authors and editors is the recently published paper by Etzerodt et al., “The Cryo-EM structure of human CD163 bound to haptoglobin-hemoglobin reveals molecular mechanisms of hemoglobin scavenging”, also in Nature Communications, two months ago. (<https://www.nature.com/articles/s41467-024-55171-4>)

As mentioned in response to reviewer 1, our manuscript was submitted before Etzerodt et al and Nature Communications has the very enlightened policy that multiple groups conducting overlapping studies is a strength rather than a concern: <https://doi.org/10.1038/s41467-020-17817-x>.

Despite the broad overlap with Etzerodt's paper, the Zhou manuscript presents additional progress in three main domains. (1) Higher-Resolution Structural Insights: The authors deliver a deeper atomic-level description of CD163–HpHb interactions, including side-chain placements and a unique visualization of the unliganded receptor's autoinhibitory arrangement. (2) Broader Ligand Range: By comparing HpHb from different isoforms, engineered Hp domains, and haptoglobin-related protein, this work elucidates the receptor's flexibility and specificity more comprehensively than the Etzerodt paper. This is important, given that an in human ligand heterogeneity exists, provided by the genetic polymorphism in haptoglobin. (3) Novel Mechanistic Angle: The new autoinhibition data explains how CD163 restricts off-target uptake, extending the fundamental biology beyond what has been described.

We thank the reviewer for highlighting features of our manuscript which differ from those described by Etzerodt. We agree that both the overlap and the differences are of interest to a reader.

Therefore, my primary suggestion is that the authors incorporate a clear comparison with Etzerodt's paper, emphasizing unique data that strongly argue for the importance and novelty of this submission. The authors should highlight the distinctions in resolution, experimental scope, and mechanistic emphasis, ensuring readers understand how their paper complements and extends the published data. The two papers provide a very complete structural explanation of hemoglobin clearance via CD163.

Our view is that, as our work was conducted entirely independently of Etzerodt et al, and completed and submitted before we knew of their findings, it is not appropriate for us to rewrite extensively. It will be of more value to see these as independent studies. Instead, we have added a paragraph to the conclusions (lines 311-326) which we think fairly notes the synergies and differences between these two manuscripts.

Reviewer #3 (Remarks to the Author):

Thank you for reviewing our paper.

Reviewer #1 (Remarks to the Author):

The revised version of the manuscript by Zhou and Higgings entitled “Scavenger receptor CD163 multimerises to allow uptake of diverse ligands” shows substantial improvements compared to the original submission. I am pleased to see that the authors have adapted the text to the longer format allowed by Nature Communications, as requested by the reviewer. The data presented are of very high quality, and the scientific question addressed is important.

We thank the reviewer for their view that the data are of high quality and the question addressed is important.

However, I still believe that—aside from personal preferences about how data should be presented and where to draw the line between interpretation and speculation—the manuscript requires further discussion/description on some findings. While the authors argue that it is not a reviewer’s role to judge the presentation of data, as long as it is technically sound, I must respectfully disagree. High-quality data is the foundation, but it is indeed a reviewer’s responsibility to assess whether it is also effectively presented.

As stated in the guidelines one of the factors to keep in mind is:

“The quality of the data — whether they are technically sound, obtained with appropriate techniques, analysed and interpreted carefully, and presented in sufficient detail.”

I will provide more specific comments below, but I would like to first highlight two major points:

1) Abstract

I had previously suggested to enrich the abstract with more specific data. The authors have chosen to retain the original version, and while I will not insist further, I believe that explicitly mentioning that the investigation involves the structural characterization of two ligation states of CD163—highlighted as a novelty over the work by Etzerodt et al. in the conclusions by the authors—would be beneficial.

Sadly, as we are at the word limit for the journal we can’t include more information into the abstract. However, we do state in lines 7-9 that dimeric and trimeric complexes are observed.

2) Conclusions

The conclusions mainly comment on the results with only limited discussion in light of existing literature. While I agree with the authors that the conclusion might focus on results, if the editorial office allows a manuscript to omit a separate “Discussion” section, then this is acceptable.

Major Points

1) Sample Preparation for Cryo-EM

While I would have approached grid preparation differently (e.g., by testing different stoichiometric ratios), I understand that revising the experimental setup retrospectively is not feasible. However, it should be clearly acknowledged that using varied stoichiometries could have helped resolve different assemblies and potentially led to a more complete understanding of the complex.

Our view is that repeating the process of cryoEM structure determination with different receptor-ligand ratios is not the correct thing to do here. One of the great powers of cryoEM is the ability to observe multiple oligomeric and conformational states from a single sample. Changing ratios might have shifted the relative occupancy of trimer and dimer states, but, with less than 8% of the collected particles ending up in the dimer and trimer classes, counting the relative proportions of particles in these classes would not be a quantitatively accurate approach. In our view, the marginal gains from this approach would not have justified the substantial deployment of publicly funded resources into expensive microscope time to conduct this study.

For the sake of reproducibility, the authors should specify not only that they used a molar excess of HpHb relative to CD163, but also the exact concentrations employed. It should be clarified if these concentrations differ from those used in SEC-MALLS, and therefore, the two experiments are not directly comparable.

We have added the quantities used to lines 361. We do not suggest that the outcomes of cryoEM and SEC-MALLS are directly comparable as one is a technique which involves averaging together images of a subset of single particles, while the other involves bulk measurements. In cryoEM, a small fraction of the particles imaged (463,222 or 456,124 from the 15,666,525 collected) are included in the particle sets which generate the maps (Extended Data Figure 2). This is standard practice. In contrast, the whole bulk sample generates the SEC-MALLS data. They should therefore not be directly compared.

Related to this, a thorough discussion of Extended Data Fig. 1c is warranted. Specifically, the authors should comment on the result that increasing the total protein concentration appears to reduce the relative abundance of high to low molecular weight species, which is not obvious considering that the stoichiometric ratio should not change. Finally, it would be highly beneficial if the authors could provide a supplementary table listing the molecular weights determined by SEC-MALLS alongside the theoretical molecular weights of the species they believe they have isolated.

The reviewer correctly points out an oversight in the legend for Extended Data Figure 1c. Although described in detail in the methods, we omitted to mention in the figure legend that the SEC-MALLS data involves measurements taken at different concentrations of CD163, but all with a constant concentration of HpHb. This explains why, as HpHb becomes increasingly occupied, the amount of free CD163 increases, as represented by the peak at ~13.5ml. We thank the reviewer for drawing our attention to this omission and have fixed it in lines 820-821.

We do not agree with the suggestion to include a table of molecular weights determined by SEC-MALLS as (i) SEC-MALLS is not sufficiently accurate to justify this and (ii) as can be seen from the slopes on the mass curves the mass varies across the SEC peaks, most likely due to variable stoichiometry. We have, however, added the theoretical mass for the dimeric and trimeric CD163:HpHb complexes to the legend, in lines 821-823, to allow readers to more easily interpret the data.

2)SPR Analysis

Two major concerns need to be addressed:

1. The authors immobilize biotinylated CD163 and its R809T variant on SPR chips to measure ligand binding affinities. However, it is unclear what assembly (monomeric, dimeric, trimeric or a mixture) they assume for the immobilized CD163. While trimers are briefly mentioned (line 176), this is not experimentally demonstrated. While it is likely that R809T is immobilized as a monomer, the authors should provide evidence that CD163 forms trimers upon immobilization or explicitly state that the oligomeric state remains uncertain. This clarification is particularly important given that the K_d values determined here differ from those reported by Etzerodt et al., who used an engineered, stable trimeric form of CD163.

We agree with the reviewer that these are important considerations in the interpretation of the SPR data and we are not aware of a method which can be used to tell whether CD163 is trimeric while conjugated to the chip surface. We are also not sure that the stable trimer of Etzerodt et al will trimerise in the same way as the native molecule, based on analysis of the lengths of the linkers they used and the structure of CD163. In our experiment, one might expect a mixture of monomers, dimers and trimers on the chip. We have added this into line 174-175. The reviewer is correct that we should not have used 'trimer' in line 178 and we have replaced this with 'complex'.

In terms of CD163 R809T, we used a low ligand density on the chip surface to attempt to minimise the likelihood that that multiple molecules were placed sufficiently close to simultaneously bind to a single ligand. This is now made clearer in the methods in lines 441-443.

2. The K_d value reported for Hp(2-2)Hb shows an uncertainty equal to the value itself. The authors should explain how this level of uncertainty affects their conclusion—that monomeric CD163 binds Hp(2-2)Hb with similar or higher affinity than trimeric CD163—a statement that is currently difficult to interpret.

The reviewer is correct in stating that the error in this data is relatively high, due to the very stable binding making it challenging to fit an accurate off-rate. The very similar shapes of these curves (see Extended Data Fig. 6a) and the overlapping dissociation constants, make us confident in our conclusion.

3)Monomeric CD163

Although the authors have made some minor changes to the narrative, the description of this mutant remains insufficient. While it is clear that the mutation introduces a glycosylation recognition site, the rationale behind its design is not. Where is the mutated residue located? Why was it chosen (e.g., based on previous structural or mutagenesis studies)? Given that this is a non-standard method for disrupting protomer-protomer interactions, the authors should clearly explain the rationale and any expected advantages.

The mutated site is N807, as stated in line 233. Its location in the multimerization interface is shown in Figure 5A.

We have added 'the structure shows that' and 'based on the structure' into lines 232-233 and 235-236 to make it clearer that the glycan insertion site was designed based on the structural information to be a location to which addition of a glycan will block complex formation.

The expected function of this glycan insertion is to 'sterically prevent multimer formation' as stated on line 236. We and others use this technique regularly as a single point mutation rarely blocks an interface, but instead has more modest effects on the affinity, while insertion of a glycan into an interface sterically prevents complex formation.

In relation to the ligand uptake the authors should state if the two receptor forms (wt and R809T) are expressed to the same levels.

The reviewer is correct to point out the importance of using cells which express equivalent levels of receptors for these comparative studies. To ensure that we were working with cells which expressed equivalent levels of CD163, we sorted the transfected cells based on fluorescence of the cytoplasmic GFP label. This is outlined in lines 491-503. We have added a few words to lines 502 and 503 to make clearer that this led to consistent and equivalent CD163 expression for wildtype and mutant receptors.

4) Role of Calcium

The calcium-binding sites are only superficially described. The authors primarily refer the reader to a supplemental table for residue identities and site positions. They should clearly indicate whether the identified sites are consistent with previously reported metal-binding motifs in SRCR family proteins and highlight any that are novel. A more detailed comment on the calcium-binding sites located at the base of the receptor would also be valuable—especially since these are mentioned in the conclusion in reference to the Etzerodt study. Additionally, the calcium concentration used is described as “physiological”, in both the rebuttal letter and the main text, however it should be acknowledged that this concentration has been previously shown (ref. 14) to be saturating.

The calcium binding sites are shown in Figure 1d and 4a. Our preference is not to reduce the readability of the manuscript for a general audience by introducing a detailed discussion of the calcium binding sites. Those particularly interested in this will be able to download the PDBs to analyse this for themselves, guided by Extended Data Table 3.

We have added reference 14 as a reference for extracellular calcium, as requested by the reviewer. It is not unexpected that CD163 will be saturated with calcium at physiological concentrations, at which it is beneficial to have strong HpHb binding.

Minor comments:

Line 46: In this context, "different ligands" should be revised to "different assemblies of the same ligand" or a similar phrase.

We prefer to retain 'different ligands' as the following sentences proceed to outline what these ligands are and includes haemoglobin as well as haptoglobin-haemoglobin.

Line 52: Add reference 27.

As described in the previous iteration of responses, we think that the appropriate way to include Etzerodt is in the conclusions, as this work was not done in the context of their study, which was published after we submitted our work. We remain of the view that this is the correct way to handle this, giving fair credit to Etzerodt et al for their discoveries, while also showing the context in which we made our discoveries. It is true that Etzerodt also showed haemoglobin uptake, but we did not know this when we conducted our study.

Line 55: Free Hb in solution is expected to predominantly exist in the dimeric form, at least in the case of modest hemolysis.

We agree with the reviewer that both dimeric and tetrameric Hb will be found in cases of hemolysis and have deleted the word 'tetramer' in this sentence.

Line 71: Cite Madsen (ref. 14) to support the choice of calcium concentration, or alternatively, cite other relevant literature.

We have added this citation.

Line 74: Add reference to Extended Data Fig. 1.

We have added this citation.

Line 104: The notation "CD163A" has not been defined previously. While its meaning is somewhat self-evident, a brief description would be beneficial.

We have added a definition.

Lines 176 and 215: The values provided refer to dissociation constants, not affinities.

We have made these changes.

Line 280: "Our structural studies of CD163 show how it mediates ligand specificity." The reviewer notes that the authors have not fully demonstrated ligand specificity at the structural level, as they have not determined the receptor's structures in complex with all relevant ligands. Rather, a combination of biochemical methods and selected structural data has been used to propose a hypothesis.

We have changed 'shows' into 'provides insight into'.

Lines 338-339: Please provide more details on the mutagenesis procedure employed.

We have clarified this.

Lines 355-357: Please clarify the meaning of "individual heads" in this context.

We have now defined this.

Legend to Extended Data Fig. 3: The same information about the blue and black parts, as presented in Extended Data Fig. 2, should be included.

We have added this.

In their rebuttal letter, the authors state: A structure with two trimeric receptors attached to the membrane, forming complexes with an HbHp dimer, is not physiologically relevant. However, prior data (Etzerodt, 2024) considered this possibility, and CD163 can also exist in a soluble form. The authors may wish to comment on this.

The reviewer makes an interesting point. However, soluble CD163 has not been studied in this manuscript, which instead is about the role of membrane-integrated CD163 in mediating HpHb uptake. Studying soluble CD163 would require measurement of its serum concentrations and extensive structural and biophysical studies, which are outside the scope of this study.

The revised version of the manuscript by Zhou and Higgins, entitled “Scavenger receptor CD163 multimerises to allow uptake of diverse ligands”, has been largely modified in line with the reviewers’ suggestions. Therefore, it can be considered for publication.

There are, however, a few minor points that the authors may want to address in the final version:

1. Regarding the large error associated with the K_d values calculated for wild-type CD163 binding to Hp(2-2)Hb, I agree with the authors that the high-affinity interaction likely hinders accurate estimation of dissociation constants. However, the K_d value for R809T CD163 binding to Hp(2-2)Hb—which is even lower—has a comparatively smaller error. The authors should clarify in the text that the large error specific to this K_d measurement limits its utility for meaningful comparison with other data.

Our view is that the reader can see this for themselves as the errors are clearly stated. The required information is therefore already provided without emphasising the point.

2. The authors note that the insertion of glycosylation sites is a strategy used by them and others to disrupt protein-protein interactions. Including a reference to a publication where this technique has been successfully applied would be highly beneficial to readers.

This is not a complex method and so there is no need to add a reference. Stick a glycan where the interface should be and the two proteins are not able to form a complex as the glycan is in the way. That is all there is to it!

3. In line 71, the authors cite Madsen et al., but do not explicitly state that the calcium concentration used in that study was confirmed to be saturating. I suggest adding this detail to clarify the experimental context.

We have added ‘and ensure calcium saturation of the receptor’.